# Bacterial cell widening alters periplasmic size and activates envelope stress responses

Matylda Zietek[1,13], Amanda Miguel[2,13], Handuo Shi [2,3,13], Iskander Khusainov [4], Abir T Asmar[5], Sri Ram [6], Morgane Wartel [1], Anna Sueki[1,7], Martin Schorb[8], Mark Goulian [9], Jean-François Collet [5,10], Martin Beck [4], Kerwyn Casey Huang [2,3,11✉] & Athanasios Typas [1,12✉]

## Abstract

The Rcs signal transduction system is a phosphorelay responsible for sensing enterobacterial cell envelope stresses. In *Escherichia coli*, the Rcs system is required to survive treatment with A22 and mecillinam, antibiotics that perturb cell size. To test whether size changes are correlated with envelope damage and thereby sensed by the Rcs system, we tuned *E. coli* cell size via A22 treatment, mutations in the cell-shape determinant MreB, and mechanically confined growth. In all conditions, cell width was strongly correlated with Rcs activation, and RcsF, the outer-membrane-localized upstream component, was essential for responding to cell width changes. Several gene deletions that induce Rcs resulted in cells that were wider than wild-type. Cryo-electron microscopy revealed that the periplasm of a wide MreB mutant is ~3 nm thinner than in wild-type cells, bringing RcsF closer to the downstream, inner-membrane-localized components of the signaling cascade. Conversely, extending the RcsF linker region in wild-type cells by ~3 nm increased Rcs activity. Thus, we propose that the Rcs system responds to changes in cell width due to altered periplasmic thickness.

**Keywords** Rcs Phosphorelay; Cell Shape; Morphogenesis; Periplasm; Gram-negative Cell Envelope
**Subject Categories** Microbiology, Virology & Host Pathogen Interaction; Signal Transduction

## Introduction

Although bacteria have the capacity to respond to environmental changes and stresses through signal transduction pathways, in many cases, how such changes and stresses are directly sensed remains unclear. In the case of defects in cell envelope integrity or assembly, damage is often sensed at the cell surface (Asmar et al, 2017). Signals are then transduced to the cytoplasm via phosphorylation to activate the expression of repair and adaptive systems, thereby maintaining viability (Hews et al, 2019). Several signaling systems have been reported to respond to stress in the Gram-negative cell envelope (Delhaye et al, 2019; Mitchell and Silhavy, 2019).

One of the most well-studied and complex signaling pathways involved in responding to envelope stress is the Rcs phosphorelay, which responds to both outer membrane (OM)- and peptidoglycan-related stresses in enterobacteria (Farris et al, 2010; Laubacher and Ades, 2008). Unlike typical two-component systems consisting of only a sensor histidine kinase and a cytoplasmic response regulator, the Rcs system has at least four additional components, including an intermediate inner membrane (IM) phosphorelay protein, an auxiliary non-phosphorylatable transcription factor, and two proteins that act upstream of the phosphorelay cascade and are associated with signal sensing (IgaA and RcsF). The OM lipoprotein RcsF, the most upstream component of the pathway, is required to activate the Rcs phosphorelay under most conditions. RcsF senses envelope damage by monitoring activity of the Bam machinery (Cho et al, 2014). In growing cells, BamA, the major component of the outer membrane porin (OMP) assembly machinery, funnels RcsF through OMPs to the cell surface (Cho et al, 2014; Rodriguez-Alonso et al, 2020). This process spatially separates RcsF from IgaA, which inhibits the Rcs pathway (Wall et al, 2018; Wall et al, 2020). The Rcs system is activated when BamA fails to bind RcsF and funnel it to OMPs, whereupon newly synthesized RcsF remains facing the periplasm and directly interacts with IgaA to activate the cascade (Cho et al, 2014; Hussein et al, 2018). RcsF can also directly sense lipopolysaccharide (LPS) defects on the OM surface via a set of charged residues (Cho et al, 2014; Farris et al, 2010; Konovalova et al, 2016; Tata et al, 2021); activation presumably occurs via RcsF release and/or immobilization facing the periplasm. RcsF signaling is sensitive to periplasmic thickness (i.e., the distance between the OM and IM):

[1]Genome Biology Unit, EMBL Heidelberg, Heidelberg 69117, Germany. [2]Department of Bioengineering, Stanford University, Stanford, CA 94305, USA. [3]Department of Microbiology and Immunology, Stanford University School of Medicine, Stanford, CA 94305, USA. [4]Department of Molecular Sociology, Max Planck Institute of Biophysics, Frankfurt, Germany. [5]de Duve Institute, UCLouvain, Brussels, Belgium. [6]School of Arts and Sciences, Azim Premji University, Bangalore, Karnataka, India. [7]Collaboration for Joint PhD Degree Between EMBL and Heidelberg University, Faculty of Biosciences, Heidelberg, Germany. [8]Electron Microscopy Core Facility, EMBL Heidelberg, Heidelberg 69117, Germany. [9]Department of Biology, University of Pennsylvania, Philadelphia, PA 19104, USA. [10]Walloon Excellence in Life Sciences and Biotechnology (WELBIO), Brussels, Belgium. [11]Chan Zuckerberg Biohub, San Francisco, CA 94158, USA. [12]Molecular Systems Biology Unit, EMBL Heidelberg, Meyerhofstraße 1, Heidelberg 69117, Germany. [13]These authors contributed equally: Matylda Zietek, Amanda Miguel, Handuo Shi. ✉E-mail: kchuang@stanford.edu; typas@embl.de

when peptidoglycan becomes untethered from the OM, RcsF cannot reach IgaA to activate the system (Asmar et al, 2017). How RcsF orientation, periplasmic thickness, and other physiological parameters such as cellular dimensions are coupled to envelope damage has yet to be determined.

A chemical genomics screen of the *E. coli* Keio library of nonessential gene knockouts (Baba et al, 2006) revealed that cells lacking components of the Rcs pathway (Δ*rcsB*, Δ*rcsD*, and Δ*rcsF*) display increased sensitivity to specific conditions that perturb the cell envelope (Nichols et al, 2011). These conditions include the MreB inhibitor A22 (Gitai et al, 2005), the PBP2 inhibitor mecillinam (Spratt, 1977), the osmolyte NaCl, and the LPS biosynthesis inhibitor CHIR-090 (Barb et al, 2007; McClerren et al, 2005). Under each of these conditions, the Rcs pathway becomes active in wild-type cells in an RcsF-dependent manner (Cho et al, 2014). During A22 and mecillinam treatment, cells also become rounder by increasing in width and decreasing in length (Tropini et al, 2014). Whether changes in cellular dimensions can lead to activation of the Rcs pathway remains unknown.

Bacterial cells possess mechanisms to directly sense local biophysical properties and architectural changes such as surface curvature (Ursell et al, 2014), envelope stress (Amir et al, 2014), and periplasmic thickness (Asmar et al, 2017). However, there is currently no bacterial signaling system known to intrinsically sense changes in cellular dimensions and/or overall shape. Cell shape, which varies across bacterial species (Young, 2006), is typically tightly maintained by a given species in a non-fluctuating environment (Cesar and Huang, 2017). Yet, changes to cell shape can have important physiological consequences, such as during infection in which smaller and rod-shaped cells are better at escaping the host immune response than larger and rounder cells (Champion and Mitragotri, 2006). Cell shape and cellular dimensions are largely controlled by cell wall and OM biosynthesis (Holtje, 1998; Silvis et al, 2021). Hence, we wondered whether the Rcs system functions through sensing of changes in global cellular dimensions.

Here, we report a general relationship between the Rcs stress response and cell width. We find that the Rcs system is activated under environmental, genetic, and mechanical perturbations to cell width. Cryo-electron microscopy of wider *E. coli* mutant cells revealed significant decreases in periplasmic thickness relative to wild-type cells, providing an explanation for increased Rcs activation by shifting RcsF closer to its IM-localized interaction partner, IgaA. Consistent with this hypothesis, in cells with wild-type periplasmic thickness, extending the RcsF linker resulted in Rcs activation. These findings support a model in which Gram-negative bacteria use RcsF to sense and respond to cell envelope rearrangements associated with cell widening.

## Results

### Rcs activation by cell-shape-perturbing antibiotics is correlated with cell width

To interrogate Rcs activation and its coupling to cellular dimensions, we used *E. coli* DH300 cells, an MG1655 Δ*argF-lac* strain carrying an *rprA*::*lacZ* transcriptional fusion in the chromosome. *rprA* is a small RNA expressed only during Rcs activation (Majdalani et al, 2002). We monitored β-galactosidase production

and cellular dimensions upon treatment with 5 µg/mL A22. Approximately 15 min after the start of A22 treatment, β-galactosidase activity started to increase in wild-type DH300, but not in Δ*rcsF* cells (Fig. 1A); note that synthesis and folding of β-galactosidase requires only a few minutes (Proshkin et al, 2010). The delayed Rcs activation coincided with the timing of changes in mean cell width, while cell length started to gradually decrease immediately after A22 treatment (Fig. 1B). Rcs activity increased in an A22 dose-dependent manner (Fig. EV1A). As expected, A22-induced changes in cell length and width were not Rcs-dependent (Fig. EV2A). Moreover, A22 treatment did not affect growth rate, at least for the first 60 min (Figs. 1A and EV1A), as previously reported (Grinnell et al, 2022; Tuson et al, 2012), although Δ*rcsF* cells were somewhat more sensitive to A22, and started to die by 2.5 h of treatment with 2 µg/mL A22 (Fig. EV2).

To concomitantly monitor Rcs activation and morphology in single cells, we transformed wild-type cells with a low-copy plasmid that expresses msfGFP from the *rprA* promoter (Methods) and performed time-lapse imaging during A22 treatment (Fig. 1C). As expected, *rprA* expression was RcsF dependent. Cell width increased more (Fig. EV2B) and at a faster rate for higher A22 concentrations (Fig. 1D). As in our population-based data, upon treatment with 5 µg/mL A22, a short (10–20 min) delay was evident before width increased and the Rcs pathway exhibited activation (Fig. 1D,E). msfGFP expression increased in an A22 dose-dependent manner (Fig. 1E), although variation in cell width within a population of cells did not consistently correlate with *rprA* expression (Fig. EV2C). These data suggest that cell-width increases may be sufficient for Rcs activation.

### Mutations to MreB cause correlated changes in cell width and Rcs activation

To test whether genetically induced cell-width increases would activate the Rcs pathway similarly to A22 treatment, we transformed *E. coli* MG1655 MreB point mutants that exhibit a range of mean cell widths (Fig. 2A) (Shi et al, 2017) with the *rprA*-msfGFP-reporter plasmid; these mutants have the same maximal growth rate as cells with the wild-type *mreB* allele (Monds et al, 2014; Shi et al, 2017). We imaged each of these strains during steady-state exponential growth and quantified msfGFP intensity using single-cell microscopy. There were no obvious correlations between cell width and length across these mutants (Fig. EV2D), consistent with previous observations (Shi et al, 2017). Mean msfGFP fluorescence was strongly correlated with cell width across a range spanning both thinner (MreB[D78V]) and wider cells than wild-type (Fig. 2B). While MreB[D78V] cells were only slightly thinner than wild-type cells, Rcs activation was significantly lower in these cells compared to wild-type (*P* < 0.001, two-tailed Student's *t* test). Deletion of *rcsF* essentially eliminated fluorescence in all strains (Fig. 2B). These results suggest that increased cell width generally activates the Rcs pathway in an RscF-dependent manner.

### Genetic perturbations that induce the Rcs pathway also affect cell width

To determine how closely cell width is connected with Rcs activation, we examined the cell dimensions of envelope mutants that are known to activate the Rcs system (Δ*opgG*, Δ*opgH*, Δ*tolB*, Δ*pgm*; Fig. 2C) (Majdalani and Gottesman, 2005). These mutants

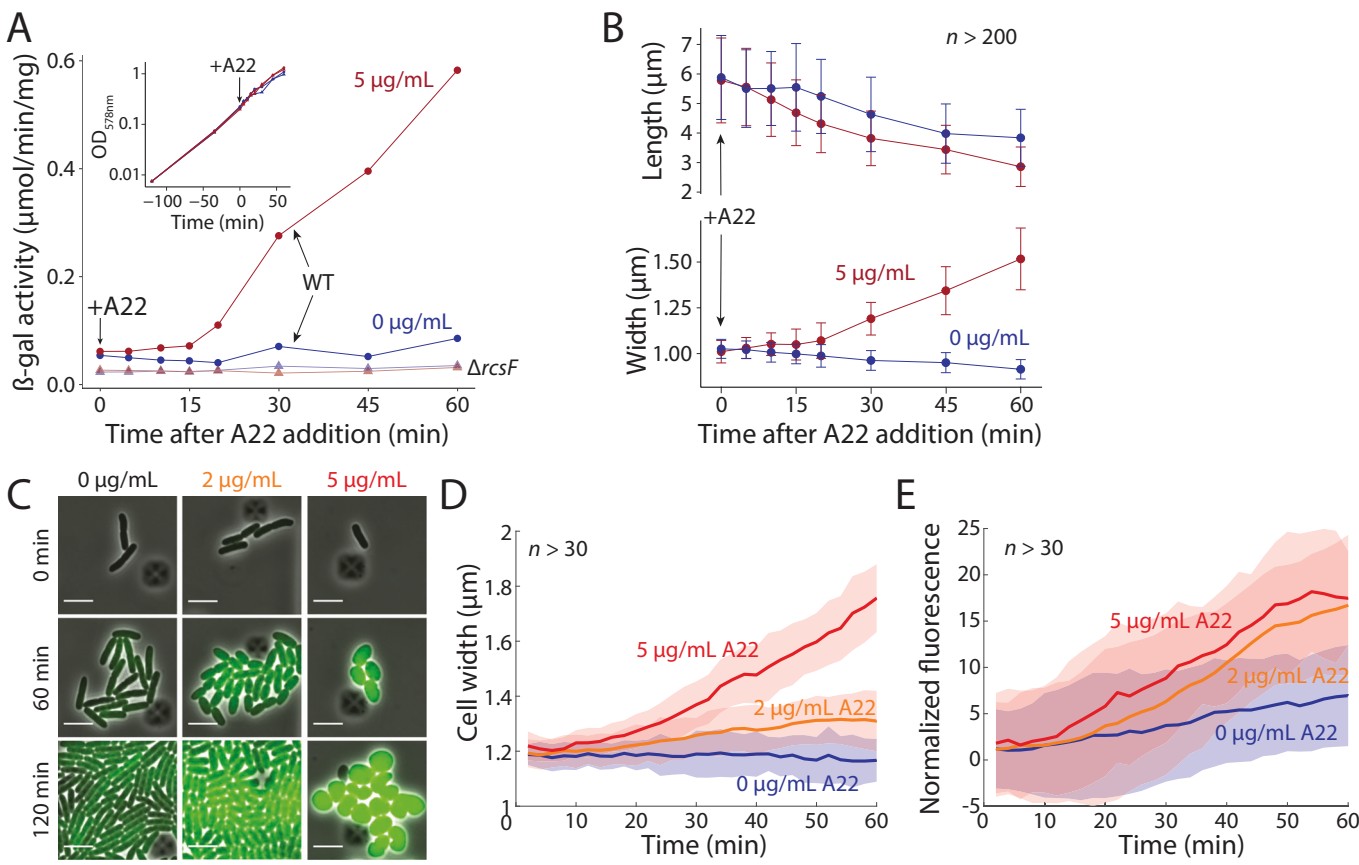

**Figure 1.    A22 concentration-dependent activation of the Rcs pathway is correlated with cell-width changes.**

(A) The Rcs system was activated ~15 min after addition of 5 μg/mL A22, as measured by the induction of chromosomal *rprA::lacZ* in a β-galactosidase assay. Cells were treated with A22 at $OD_{578 nm} = 0.3$. Inset: growth was unaffected by A22 addition. The Rcs system was not activated in Δ*rcsF* cells (triangles). Data are representative of four biological replicates (see Fig. EV1B for other replicates). (B) Cell length gradually decreased starting immediately after A22 addition, while width started to increase ~15 min after A22 addition, a similar delay as for Rcs activation in (A). Curves are means and error bars represent 1 standard deviation (SD, *n* > 200 cells for every time point). Cells were sampled and fixed from the same experiment shown in (A). (C) Time-lapse images of typical *E. coli* cells expressing msfGFP from the Rcs-dependent *rprA* promoter on plasmid pMZ13 ("Methods") treated with various concentrations of A22. Scale bars: 5 μm. (D) Median cell width of the populations represented in (C) increased over time, at a faster rate for higher A22 concentration. Curves are medians and shaded regions represent 1 SD estimated from the median absolute deviation. (*n* > 30 cells for every time point). (E) Rcs activation increased in an A22 dose-dependent manner. Curves are medians of msfGFP fluorescence normalized by cell volume, and shaded regions represent 1 SD estimated from the median absolute deviation. (*n* > 30 cells for every time point, and during the interval between 50 and 60 min, $P < 10^{-9}$ for induction levels between any pairwise comparison of 0, 2, and 5 μg/mL A22, two-tailed Student's *t* tests). Even untreated cells exhibited a basal level of msfGFP production as cell density increased. Source data are available online for this figure.

perturb the cell envelope in different ways, with *opgG* and *opgH* encoding osmo-regulated periplasmic glucans (Bohin and Kennedy, 1984), *tolB* encoding a key component of the Tol-Pal trans-envelope machinery that regulates OM constriction and home-ostasis (Connolley et al, 2021; Szczepaniak et al, 2020; Tan and Chng, 2022), and *pgm* encoding a phosphoglucomutase that has been implicated in cell division (Hill et al, 2012; Lu and Kleckner, 1994). Each of these mutants had a larger mean cell width than wild-type (Fig. 2C). Moreover, Rcs activation was dependent on RcsF and scaled with width changes (Fig. 2D), supporting a general connection between width and Rcs activation.

## Mechanical deformation leads to Rcs activation

In a previous study, *E. coli* cells placed under mechanical confinement via a compressing membrane gradually grew into

pancake shapes and increased in cell width (as measured in the imaging plane) (Si et al, 2015). To test whether such mechanically induced width changes would activate RcsF similarly to chemical and genetic perturbations, we subjected wild-type and Δ*rcsF* cells carrying the *rprA*::GFP-reporter to mechanical confinement (Fig. 3A). Cells grew into a wide range of morphologies due to local changes in the degree of confinement in our device, and wild-type cells with larger width typically exhibited high fluorescence, while even very wide Δ*rcsF* cells did not express GFP (Fig. 3B). In some clusters of cells with a combination of wide and thin cells, the thin cells exhibited high fluorescence, likely due to high induction in the parent cells from which the thin cells originated. None-theless, Rcs activation in wild-type cells was correlated with the minor axis of cells (a proxy for width) (Fig. 3C). Thus, mechanically induced width changes also result in RcsF-dependent Rcs activation.

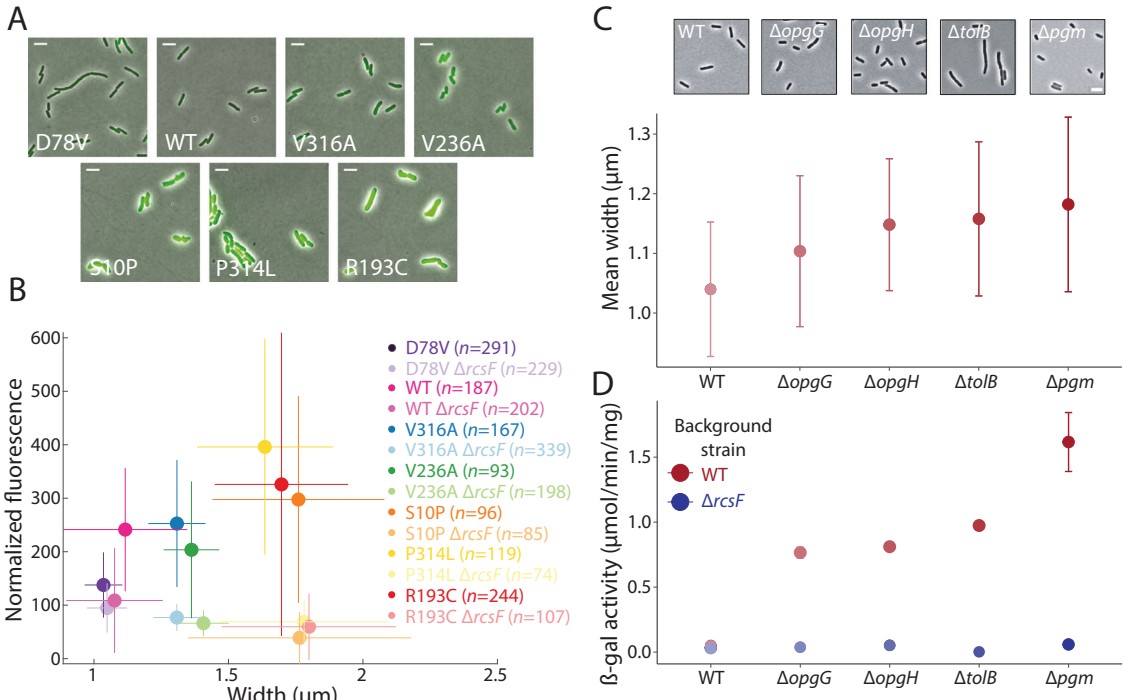

**Figure 2.  Rcs activation is correlated with cell width in mutants with altered cell size.**

(A) Overlay of phase contrast and msfGFP fluorescence from the Rcs-dependent *rprA* promoter in MreB mutants with a range of cell widths. WT: wild-type. Scale bars: 5 μm. (B) msfGFP intensity normalized to cell volume was highly correlated with cell width across the MreB mutants in (A) (Pearson's $r = 0.88$, $P = 0.009$, two-tailed Student's *t* test). Fluorescence remained low in all mutants when *rcsF* was deleted. Data points are mean values, and error bars represent 1 SD. The number of cells analyzed is denoted in the legend for each strain. (C) Representative images (top) and mean cell width (bottom) of cells from various single-gene deletion mutants known to activate the Rcs system. Cells were harvested during exponential growth in LB and fixed. Circles are means and error bars represent 1 SD ($n > 1100$ cells per strain). All mutants have significantly larger cell width compared to WT ($P < 10^{-8}$, two-tailed Student's *t* tests). Scale bar: 5 μm. (D) Rcs pathway activity as measured by β-galactosidase of the *rprA::lacZ* fusion in the same mutants and conditions as in (C). Rcs activity was higher in wider deletion mutants. Circles are means and error bars represent 1 SD. ($n \geq 3$ replicates per strain). The error bars for all strains besides Δ*pgm* are smaller than the circles. For each strain, deletion of *rcsF* resulted in significantly lower Rcs pathway activity ($P = 10^{-4}$, $10^{-5}$, $10^{-6}$, $10^{-7}$, and $3 \times 10^{-4}$, for WT and four mutants, respectively; two-tailed Student's *t* tests). All four mutants had higher Rcs pathway activities than WT ($P < 10^{-6}$, two-tailed Student's *t* tests). Source data are available online for this figure.

## A wider MreB mutant exhibits decreases in periplasmic thickness sufficient to activate the Rcs pathway

To probe the mechanism of Rcs activation in wider MreB mutants, we first tested whether RcsF transport from the IM to the OM was disrupted. RcsF remaining in the IM leads to constitutive activation of the Rcs system (Cho et al, 2014; Tao et al, 2012). However, RcsF localization to the OM was intact in the widest mutant MreB^R193C (Fig. EV3). A previous study showed that transmission of damage signals from the cell envelope by RcsF, located in OM, to the IM-localized IgaA depends on the distance between the two membranes relative to the length of the RcsF flexible linker (Asmar et al, 2017). We thus wondered whether cell widening alters periplasmic thickness in a manner that allows RcsF to reach IgaA more readily and activate the Rcs system.

Since the Rcs pathway was highly active in wide MreB mutants during normal growth (Fig. 2B), we sought to determine whether MreB^R193C cells had different periplasmic thickness from wild-type cells. Using cryo-electron microscopy, we acquired high-resolution images of wild-type and MreB^R193C cells. We focused on the midsection of cells, which exhibited more uniform envelope architecture (Fig. 4Ai), computationally segmented the IM and

OM (Fig. 4Aii,iii; "Methods"), and measured the intermembrane distance at regularly spaced points as a proxy for periplasmic thickness. (Figs. 4Aiv and EV4). The wider MreB^R193C mutant exhibited a significant decrease in periplasmic thickness by ~3 nm (Figs. 4B and EV5). This decrease in periplasmic thickness should result in RcsF that reaches the OM being closer to the IM-localized IgaA, facilitating their interaction. Consistent with this idea, extending the RcsF linker by 7 amino acids (equivalent to 2–3 nm) in wild-type cells resulted in Rcs activation (Fig. 5A). This RcsF^+7 construct retained its outer membrane localization (Fig. 5B), demonstrating that the increase of RcsF length relative to the intermembrane distance in MreB^R193C cells led to Rcs activation. Thus, Rcs activation during cell widening may be caused by changes in periplasmic thickness.

## Discussion

Our data support a strong correlation between width and Rcs activation, but does cell width directly activate the Rcs pathway? In the case of mechanical confinement, the width variation across the population due to heterogeneity in the application of force was

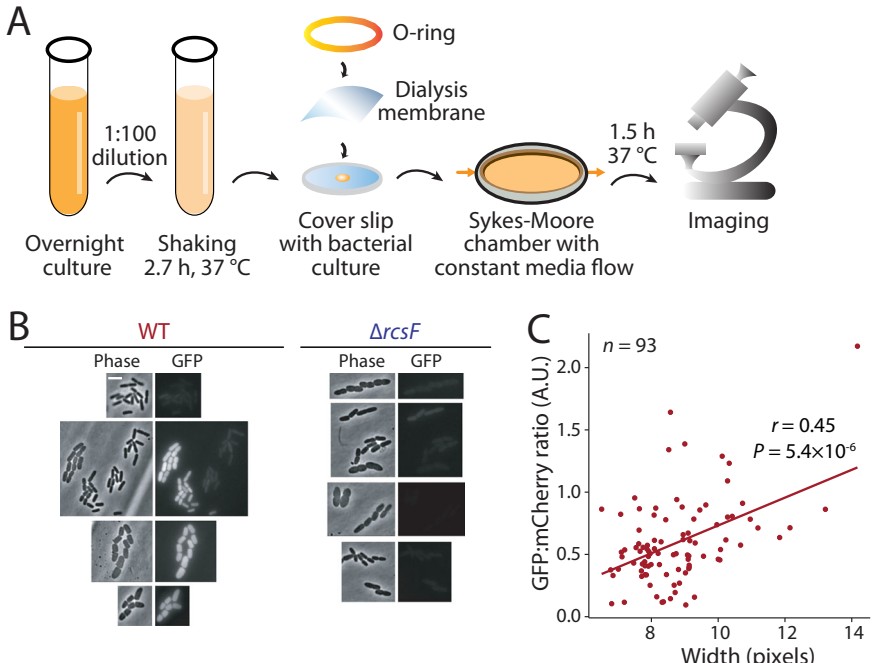

**Figure 3. Mechanical perturbations activate the Rcs pathway.**

(A) Schematic of protocol for monitoring cell growth under mechanical confinement. Log-phase cells carrying an *rprA*::GFP fusion on plasmid pP_rprA-gfp and constitutively expressing mCherry from the chromosome were placed under a permeable membrane, and pulling forces were exerted on the membrane edge with a Sykes–Moore chamber to reduce the volume under the membrane. In these conditions, cells grew in different shapes depending on the differential confinement forces exerted across the membrane surface. Cells were imaged after 1.5 h. (B) Representative images of an experiment as described in (A). Upon mechanical confinement, wild-type and Δ*rcsF* cells increased similarly in major- and minor-axis lengths (proxies for cell length and width, respectively). On the other hand, GFP intensity increased in wider wild-type (WT) cells, but not in any Δ*rcsF* cells. Scale bar: 5 μm. (C) The ratio of GFP to mCherry intensity was correlated with cell width in wild-type cells. *r*: Pearson's correlation coefficient. *P* value is from a two-tailed Student's *t* test. Source data are available online for this figure.

correlated with *rprA* expression (Fig. 3C), consistent with the results of a study published while this study paper was under review showing that mechanical compression can induce the Rcs system (Mason et al, 2023). *rprA* expression also scaled with mean cell width across A22 concentrations (Fig. 1D,E) and cell-width mutants (Fig. 2B,D), although within-population variability in width at fixed A22 concentration was not consistently correlated with *rprA* expression (Fig. EV2C), potentially due to the relatively minor fluctuations of cell width and fluorescence signal reporting on Rcs activation with some delay. However, despite the many conditions reported here in which cell width and Rcs activation were strongly correlated, the Rcs system can also be activated in conditions in which cell width is unaltered. Defects in LPS can transiently activate the Rcs system (Farris et al, 2010; Konovalova et al, 2016), and Δ*rcsF* cells are sensitive to CHIR-090 treatment (Nichols et al, 2011) even though CHIR-090 activates the Rcs pathway in wild-type cells but does not change cell width (Fig. EV6A). Deletion of *rfaF*, which encodes an enzyme involved in LPS core biosynthesis (Brabetz et al, 1997), resulted in Rcs activation without affecting cell width (Fig. EV6B).

Taken together, width changes are sufficient but not necessary for Rcs activation, suggesting that width is correlated with another feature(s) that ultimately dictates Rcs activation. Our findings implicate periplasmic thickness as a key factor in Rcs activation, which was previously shown to be affected by the presence and length of the lipoprotein Lpp and to dictate communication across

the envelope by RcsF to IgaA (Asmar et al, 2017). Extension of the RcsF linker has been shown to compensate for increases in periplasm thickness in *lpp* mutants (Asmar et al, 2017). Here, extension of the RcsF linker by 2–3 nm was sufficient to activate the Rcs system in wild-type cells (Fig. 5A), to a similar extent (four-fold to five-fold) as Rcs was activated in MreB[R193C] cells (Fig. 2B). Extension of the linker or decrease of periplasmic thickness brings RcsF (3100 copies/cell) closer to the integral IM protein IgaA (~220 copies/cell) (Li et al, 2014), with which RcsF binds with high affinity ($K_D \sim 1.6$ nM) and of which it is in excess (Dekoninck et al, 2020). Consistent with the hypothesis that the Rcs system is activated by a mismatch between RcsF size and periplasmic thickness, our cryo-electron microscopy data indicated that the periplasm of MreB[R193C] cells is ~3 nm thinner than in wild-type cells (Fig. 3B). For mutants such as Δ*rfaA* (Fig. EV6B), expression of RcsF[+7] could be used in the future to query whether alteration of periplasmic size explains the activation of Rcs without changes in cell width.

How cell shape and size are connected to periplasmic dimensions remains mysterious. One possibility is that cell widening is correlated with changes in turgor that cause the periplasm to become thinner. Unfortunately, periplasmic size is essentially unknown in all but a few cases. Cryo-EM provides a means to quantify periplasmic dimensions in a native state, but our measurements highlight the high variability in periplasmic thickness both across cells and within cells along the cell surface (Figs. 4B and EV5). This heterogeneity in periplasmic thickness, which has previously been underappreciated (Asmar et al, 2017),

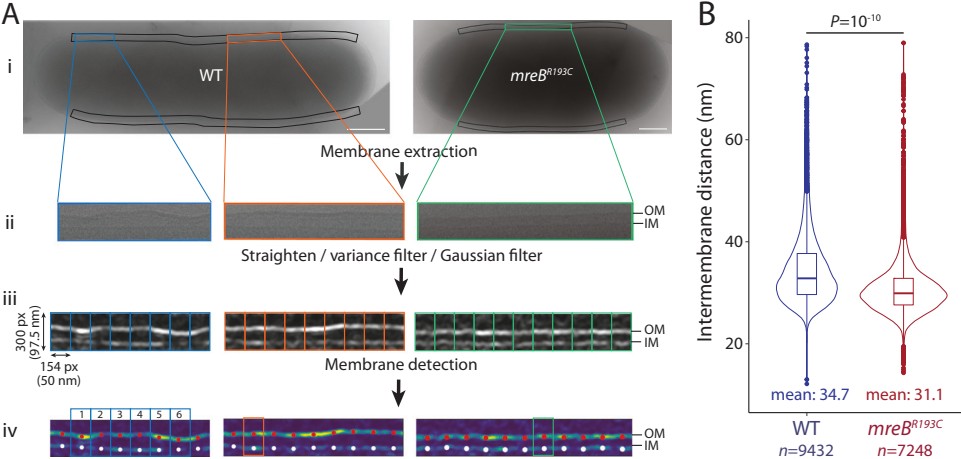

**Figure 4. Periplasmic thickness is decreased in wider *mreB^R193C* cells compared with wild-type.**

(A) Blended montage of a stack of cryo-EM images of representative cells with average intermembrane distance for wild-type (left) and *mreB^R193C* (right). Measurements were carried out along cylindrical regions (black boxes in (i)) excluding the poles. Both wild-type and *mreB^R193C* cells exhibited diverse membrane morphologies from straight (orange, green) to wrinkled (cyan). The processing workflow involved membrane extraction (i), membrane enhancement via straightening and filtering (ii), and separation into 50-nm segments (iii). Measurements were computed as the distance between the red and white dots in each segment (Fig. EV4). OM, outer membrane; IM, inner membrane. Scale bars: 500 nm. (B) Intermembrane distances were lower on average in *mreB^R193C* cells compared with wild-type. Violin plot shows the smoothened density of measurements. *n* represents the number of points at which measurements were made, for 86 wild-type and 71 *mreB^R193C* cells pooled from three biological replicates. *P* value is from a two-tailed Student's *t* test. Box plots show the median and first and third quartiles, and the whiskers are 1.5 interquartile ranges from the corresponding quartiles. Data points outside the whiskers are plotted as individual points. Source data are available online for this figure.

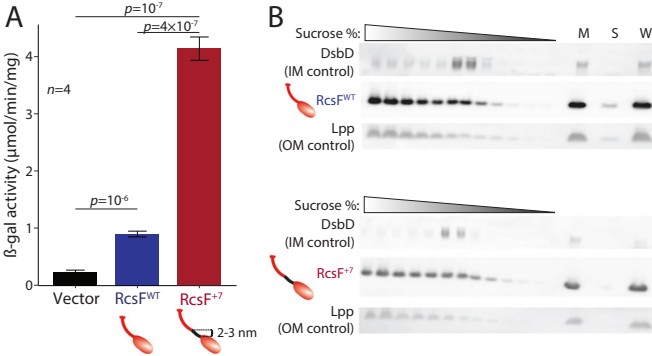

**Figure 5. Extension of the RcsF linker results in Rcs activation in wild-type cells without affecting RcsF localization.**

(A) Activation of the Rcs pathway as measured by β-galactosidase activity was higher in a mutant expressing RcsF with its linker region extended by seven amino acids (RcsF^+7) cells compared to wild-type or the vector control in Δ*rcsF* cells. Bacteria were harvested for a β-galactosidase assay at $OD_{600\,nm} = 0.6$. *P* values shown are from two-tailed Student's *t* tests, *n* = 4 replicates each. Data points are mean±1 SD. (B) Sucrose-gradient fractionation of cells expressing RcsF^WT or RcsF^+7 showed similar localization of both proteins to the outer membrane (OM). Immunoblots of RcsF and controls (DsbD for the inner membrane (IM) and Lpp for the OM). M total membrane sample prior to fractionation, S soluble non-membrane fraction, W whole-cell lysate. Source data are available online for this figure.

suggests that there may be the potential for Rcs activation in every cell dependent on the localization of RcsF and its associated machinery, and motivates the development of methods that can quantify periplasmic dimensions at the poles (which are difficult to discern without tomography) and to directly measure Rcs

activation at a subcellular level. Importantly, while our data suggest that cell width and periplasmic thickness are correlated, defects in the Bam machinery and LPS can also activate the Rcs system, the latter without affecting periplasmic size. Newly transported RcsF in the OM is ushered to the cell surface by the Bam machinery, and interactions with BamA ($K_D \sim 400$ nM; 3900 copies/cell) (Li et al, 2014; Rodriguez-Alonso et al, 2020) and with the highly abundant OmpA ($K_D \sim 100$ μM; 100,000–200,000 copies/cell) (Dekoninck et al, 2020; Koebnik et al, 2000; Li et al, 2014) prevent RcsF from reaching to IgaA and activating the Rcs signaling system.

Independent of the mechanism, the role of Rcs activation upon changes to cellular dimensions is intriguing. Periplasmic glucans, which are induced by the Rcs pathway (Majdalani and Gottesman, 2005), may also physically impact periplasmic thickness, as deletion of the genes responsible for their production activates Rcs (Ebel et al, 1997) and increases cell width (Fig. 2D). Rcs activation as cells widen may serve to protect from increased incidence of envelope stress (e.g., protein misfolding and mislocalization (Majdalani and Gottesman, 2005; Wall et al, 2018)) and/or greater sensitivity to rupture due to changes in the distribution of mechanical stresses in the envelope, and may be involved in width-dependent feedback on *mreB* transcription (Silvis et al, 2021). In particular, Rcs-induced production of colanic acid (Gervais et al, 1992) could create an additional layer of mechanical strength to protect the cells from rupture, similar to how the OM contributes to sustaining the internal turgor pressure (Rojas et al, 2018). Finally, given the recently discovered connections between cell width and length through surface area-to-volume ratio (Harris and Theriot, 2016; Shi et al, 2021), the width dependence of Rcs activation suggests that the Rcs pathway may play an important role in cell division and length determination. Interestingly, the Rcs response has been

reported to directly activate *ftsAZ* expression (Carballes et al, 1999), is required for de novo envelope biogenesis (Ranjit and Young, 2013), and leads to non-growing cells upon constitutive activation (*igaA* depletion (Cho et al, 2014; Dominguez-Bernal et al, 2004)). In an accompanying paper (Miguel et al, 2025), we uncouple cell-shape sensing of the Rcs system from its response and characterize the effects of ectopic and constitutive Rcs activation on cell growth and shape. Together, these studies highlight the intimate connections between cellular structure and stress sensing.

# Methods

### Reagents and tools table

| Reagent/resource | Reference or source | Identifier or catalog number |
|---|---|---|
| **Experimental models** | | |
| *E. coli* Δ*rcsF::cat* | Mori laboratory (Nara, Japan) | Aska collection |
| *E. coli* *rprA::lacZ* MG1655 (*argF-lac*)U169 used as WT | Majdalani et al, 2002 | DH300 |
| *E. coli* DH300 Δ*pgm*::kan | This study | CAG60488 |
| *E. coli* Δ*rcsF::kan*, Δ*opgG::kan*, Δ*opgH::kan*, Δ*tolB::kan*, Δ*pgm::kan* | Baba et al, 2006 | Keio collection of gene deletions |
| *E. coli* F- lambda- *ilvG- rfb*-50 *rph*-1 | Oldewurtel et al, 2021 | MG1655 |
| *E. coli* MG1655 Δ*xylA* Δ*xylFG::P_tetA::mCherry* | Roggiani and Goulian, 2015 | MMR60 |
| *E. coli* DH300 Δ*rcsF::cat* Δ*opgG::kan* | This study | NT1032 |
| *E. coli* DH300 Δ*rcsF::cat* Δ*opgH::kan* | This study | NT1033 |
| *E. coli* DH300 Δ*rcsF::cat* Δ*tolB::kan* | This study | NT1035 |
| *E. coli* DH300 Δ*rcsF::cat* Δ*pgm::kan* | This study | NT1036 |
| *E. coli* DH300 Δ*opgG::kan* | This study | NT1038 |
| *E. coli* DH300 Δ*opgH::kan* | This study | NT1039 |
| *E. coli* DH300 Δ*tolB::kan* | This study | NT1041 |
| *E. coli* DH300 Δ*rcsF::kan* | This study | NT4043 |
| *E. coli* MG1655 Δ*mreBCD* | Shi et al, 2017 | RDM893 |
| *E. coli* RDM893 pMreB^WT | Shi et al, 2017 | NT4151 (MreB^WT) |
| *E. coli* RDM893 pMreB^R193C | Shi et al, 2017 | NT4157 (MreB^R193C) |
| *E. coli* RDM893 Δ*rcsF* | This study | KCH1453 |
| **Recombinant DNA** | | |
| N/A | | |
| **Antibodies** | | |
| Horseradish peroxidase-conjugated goat anti-rabbit IgG | Sigma | Cat. #F9887 |

| Reagent/resource | Reference or source | Identifier or catalog number |
|---|---|---|
| **Oligonucleotides and other sequence-based reagents** | | |
| Plasmid pAM238 | Gil and Bouche, 1991 | IPTG-regulated P_{lac}, pSC101-based, spectinomycin resistance |
| Plasmid pAC581 | Clarke and Voigt, 2011 | |
| Plasmid pMZ13 | This study | *msfGFP* under *rprA* promoter, pAC581-based, chloramphenicol resistance |
| Plasmid pSC202 (pRcsF^WT) | Cho et al, 2014 | pAM238 with *rcsF^WT* |
| Plasmid pRcsF^{+7} | Asmar et al, 2017 | pAM238 with *rcsF^{+7}* |
| Plasmid pMreB^WT | Shi et al, 2017 | |
| Plasmid pMreB^D78V | Shi et al, 2017 | |
| Plasmid pMreB^V316A | Shi et al, 2017 | |
| Plasmid pMreB^V236A | Shi et al, 2017 | |
| Plasmid pMreB^P314L | Shi et al, 2017 | |
| Plasmid pMreB^S10P | Shi et al, 2017 | |
| Plasmid pMreB^R193C | Shi et al, 2017 | |
| **Chemicals, enzymes, and other reagents** | | |
| N/A | | |
| **Software** | | |
| Morphometrics | Ursell et al, 2017 | |
| **Other** | | |
| N/A | | |

## Methods and protocols

### Strains and plasmids

All bacterial strains and plasmids used in this study are listed in the Reagents and Tools Table. The *E. coli* K12 strain MG1655 derivative, DH300 (Majdalani et al, 2002), was used as the main genetic background unless otherwise mentioned, and is referred to as wild-type. Rcs activity was monitored by either β-galactosidase assays using a chromosomal transcriptional fusion of the Rcs-dependent promoter of *rprA* to *lacZ* (Majdalani et al, 2002), or by fluorescence microscopy using a plasmid expressing P*rprA::sfGFP* (pMZ13). Gene deletion strains (Δ*rcsF::kan*, Δ*opgG::kan*, Δ*opgH::-kan*, Δ*tolB::kan*, Δ*pgm::kan*) were created by P1 transduction of the corresponding Keio library mutant (Baba et al, 2006) into the DH300 background. All deletions were verified by PCR/Sanger sequencing. When necessary, antibiotic markers were flipped out (Datsenko and Wanner, 2000). MreB mutants were created in MG1655 as previously described (Shi et al, 2017). Deletion of *rcsF* from the MreB mutants was accomplished via P1 transduction

from the Δ*rcsF* Keio strain. pRcsF^WT and pRcsF^+7 plasmids were described in (Asmar et al, 2017).

The pMZ13 plasmid was constructed via Gibson assembly from pP_rprA-gfp, a derivative of pAC581 (Clarke and Voigt, 2011), which has copy number ~15 and contains codon-optimized *gfpmut3* expressed from the *rprA* promoter. GFPmut3 was replaced with monomeric super-folder GFP (*E. coli* codon-optimized with the V206K mutation), which was PCR-amplified from pSFGFP-N1 (Addgene) (Pedelacq et al, 2006).

The strain (MMR60) used for mechanical confinement contains the pP_rprA-gfp plasmid and *mcherry* expressed from the chromosome for normalization of the GFP signal (Roggiani and Goulian, 2015).

### Growth conditions

Unless otherwise stated, for single-cell time-lapse imaging, cells were grown in LB (Lennox formulation) at 37 °C overnight, then diluted 1:200 and grown for 1.5 h. When appropriate, drugs were added to the culture and cells were incubated for an additional 30 min. Cells were then placed on agar pads and imaged for 60 min.

For probing *rprA::lacZ* activity in knockout strains (Fig. 2D), cells were grown overnight in LB at 37 °C, then diluted to an $OD_{578\,nm}$ of 0.0075. Rcs system activity was measured at $OD_{578\,nm} = 0.5$ by β-galactosidase assay using a standard protocol (Miller, 1972). For A22 experiments (Figs. 1A and EV1), cells were grown in LB (Lennox) at 37 °C overnight, then diluted to an $OD_{578\,nm}$ of 0.0075. A22 was added when $OD_{578\,nm}$ reached 0.1 or 0.3, and Rcs system activity was monitored for 60 min by measuring β-galactosidase activity.

For comparing RcsF^WT and RcsF^+7 activity (Fig. 5A), β-galactosidase assays were performed as described in (Asmar et al, 2017).

### Single-cell imaging

For batch-culture experiments, cellular dimensions were examined in parallel with *rprA* promoter activity. For each time point, 1 mL of cell culture was harvested. Cells were fixed by incubation with 4% formaldehyde for 10 min at room temperature and centrifuged for 5 min at $1000 \times g$ before being washed three times with 1 mL of PBS. Next, cells were immobilized on 1% agarose pads and imaged with a Nikon Eclipse Ti inverted microscope, equipped with a Nikon DS-Qi2 camera and a Nikon Plan Apo Lambda 60X oil Ph3 DM phase-contrast objective. Images were acquired with NIS-Elements v. AR4.50.00 and analyzed using *Morphometrics* (Ursell et al, 2017) and custom Matlab scripts.

For experiments examining cell width and *rprA::msfGFP* expression under A22 treatment (Fig. EV2), cells were diluted 1:5000 from an overnight culture. At $OD_{600\,nm} = 0.4$, cells were diluted 1:200 in 0, 0.25, 0.5, 1, or 2 µg/ml A22 and incubated for 2.5 h before imaging on an agarose pad.

For time-lapse experiments under A22 treatment (Fig. 1C–E), cells were back-diluted 1:5000 from an overnight culture. At $OD_{600\,nm} = 0.2$, cells were back-diluted 1:10 onto LB pads made with 1% agarose and 0, 2, or 5 µg/ml A22. Cells were imaged under phase contrast and fluorescence for 2 h.

### Image analysis

Images were analyzed using *Morphometrics* (Ursell et al, 2017) and custom Matlab scripts. Time-lapse data were preprocessed using the machine learning segmentation software *DeepCell* (Van Valen et al, 2016) with manually curated training datasets specific to the microscope used for imaging. The contour outputs from *DeepCell* were then processed using *Morphometrics*.

### Microfluidics

For single-cell tracking in CellAsic microfluidic flow cells, cells were diluted 1:500 and grown for 3.5 h. Cells were then diluted to $OD_{600\,nm} \sim 0.001$ and placed in a CellAsic B04A chamber. Once introduced into the imaging chambers, cells were allowed to grow in LB for 20 min before switching to LB supplemented with A22 and were imaged for >4 h.

### Mechanical confinement using a Sykes–Moore chamber

An overnight culture of *rprA*-GFP cells grown in Minimal A medium with 0.2% glucose, 1 mM $MgSO_4$, and 20 µg/mL chloramphenicol was diluted 100-fold and placed in a roller drum at 37 °C for 2.7 h. Five microliters of this culture were pipetted onto a 25-mm round cover slip and then covered with a cellulose acetate dialysis membrane. The membrane had been wetted in distilled water and dried with a filter paper, which exerts a confining pressure due to the hydrophobic effect. An O-ring was placed on the membrane, and the entire setup was secured in a Sykes–Moore chamber (Bellco Glass, Cat. #1943-11111). The chamber was then filled with Minimal A medium with 0.2% glucose, 1 mM $MgSO_4$, and 20 µg/mL chloramphenicol. Imaging was performed in a temperature-controlled enclosure set to 37 °C. Phase contrast and GFP images were acquired after 1.2 h of growth in the chamber. The microscope and image acquisition were essentially as previously described (Libby et al, 2010).

### Sucrose density fractionation

For strains carrying MreB^WT and MreB^R193C, inner and outer membranes were separated using a sucrose density gradient as previously described (Anwari et al, 2010; Sueki et al, 2020). For DH300 RcsF^WT and RcsF^+7 strains, cell fractionation was adapted from (Cho et al, 2014). Four hundred milliliters of cell culture were grown until $OD_{600\,nm} = 0.6$. Cells were harvested via centrifugation at $6400 \times g$ and 4 °C for 15 min, washed with TE buffer (50 mM Tris-Cl [pH 7.8], 1 mM EDTA), and resuspended in 20 mL of TE buffer. One milligram of DNase I (Roche), 1 mg of RNase A (Thermo Scientific) and a tablet of Protease Inhibitor Cocktails (Roche cOmplete™) were added to cell suspensions, and cells were passed through a French pressure cell at 12,000 psi. After adding $MgCl_2$ to a final concentration of 2 mM, the lysate was centrifuged at $4200 \times g$ and 4 °C for 8 min to remove cell debris. Then, 16 mL of the supernatant was placed on top of a two-step sucrose gradient (2.3 mL of 2.02 M sucrose in 10 mM HEPES [pH 7.5], 6.6 ml of 0.77 M sucrose in 10 mM HEPES [pH 7.5]). The samples were centrifuged at $130,000 \times g$ for 3 h at 4 °C in a 55.2Ti Beckman rotor. After centrifugation, the soluble and membrane fractions (12 mL) were collected. The membrane fraction was diluted two times with 10 mM HEPES [pH 7.5]. To separate the membranes, 7 mL of the diluted membrane fraction were loaded on top of a second sucrose gradient (10.5 mL of 2.02 M sucrose, 12.5 ml of 1.44 M sucrose, 7 ml of 0.77 M sucrose, all in 10 mM HEPES [pH 7.5]). Samples were then centrifuged at $82,000 \times g$ for 16 h at 10 °C in a SW 28 Beckman rotor. Approximately 30 fractions of 1.5 mL were collected, and odd-numbered fractions were loaded on SDS-PAGE gels, transferred onto a nitrocellulose membrane, and probed with specific antibodies.

## Immunoblotting

To visualize MreB$^{WT}$ and MreB$^{R193C}$ fractionation, protein samples were separated by SDS-PAGE and transferred onto PVDF membranes (IMMOBILON P). To visualize DH300 RcsF$^{WT}$ and DH300 RcsF$^{+7}$ fractionation, protein samples were separated in 4–12% SDS-PAGE gels (Life Technologies) and transferred onto nitrocellulose membranes (GE Healthcare Life Sciences). The membranes were then blocked with 5% skim milk in 50 mM Tris-HCl [pH 7.6], 0.15 M NaCl, and 0.1% Tween20 (TBS-T). TBS-T was used in all subsequent steps of the immunoblotting procedure. Anti-BamA (1:10,000, gift from Lithgow lab at Monash University, raised against the soluble BamA POTRA domains) (Gunasinghe et al, 2018), anti-SecG (1:6000, gift from Tokuda lab at University of Morioka, Japan), anti-RcsF (1:20,000) (Leverrier et al, 2011), anti-DsbD (1:2000) (Stewart et al, 1999) and anti-Lpp (1:7000) (Asmar et al, 2017) rabbit antisera were used as primary antibodies. The membranes were incubated with horseradish peroxidase-conjugated goat anti-rabbit IgG (Sigma) at a 1:5000 dilution. Labeled proteins were detected via chemiluminescence (Pierce ECL Western Blotting Substrate, Thermo Scientific) and exposed on X-ray films (Kodak Biomax MR-1) or visualized using a GE ImageQuant LAS4000 camera (GE Healthcare Life Sciences).

## Cryo-electron microscopy

MreB$^{WT}$ and MreB$^{R193C}$ strains were grown in LB overnight at 37 °C, diluted to an OD$_{578 nm}$ of 0.0075, and grown in LB at 37 °C until an OD$_{578 nm}$ of 0.2. Cells were collected by centrifugation for 5 min at $1000 \times g$ at 22 °C and concentrated to an OD$_{578 nm}$ of 30 in fresh LB. Concentrated cells (3.5 μL) were immediately applied on glow-discharged (two cycles of 45 s at 15 mA) C-flat$^{TM}$ 4/1 grids (Photochips, Inc). Cells were plunge-frozen using a Vitrobot Mark IV (Thermo Fisher) with a wait time of 0 s, blot time of 3 s, blot force of 3, and drain time of 0 s at constant 100% humidity and at 22 °C. Transmission electron microscopy (TEM) images were collected at the EMBL Electron Microscopy Core Facility using a 200 keV FEI Talos Arctica TEM (Thermo Fisher) equipped with an autoloader and Falcon II direct electron detector (Thermo Fisher) at a pixel size of 0.3266 nm (nominal magnification 45,000X), and a defocus of −10 μm. To image entire cells at high magnification, projection images were collected as a montage of stacks of the field of view in Serial EM software (Mastronarde, 2005).

## Intermembrane distance measurement

Montaged images were pre-blended in etomo (Kremer et al, 1996), and the edges were fixed manually in MIDAS (Kremer et al, 1996). Membranes were segmented in FIJI (Schindelin et al, 2012) after application of a 3-nm Gaussian filter and 10-nm variance filter. Intermembrane distances were measured using a custom Python script that calculates the distance between two major peaks in the image using a gray-scale gradient. This calculation results in values corresponding to the distance between the centers of the inner and outer membranes, which we refer to as intermembrane distance. To account for spatial autocorrelation, each intermembrane distance calculation was an average over a 50-nm segment. The poles of the cells were excluded. The outcomes of our analysis were manually validated, and clear miscalculations were removed from the final histograms (Fig. EV4B). In total, ~30 cells from each strain were quantified for each of the three biological replicates.

# Data availability

Scripts for cryo-electron microscopy image analysis are available at the repository https://github.com/martinschorb/membranedist.

The source data of this paper are collected in the following database record: biostudies:S-SCDT-10_1038-S44318-025-00534-w.

# Peer review information

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

## Acknowledgements

The authors thank the Huang and Typas labs for useful discussions. This work was supported by a National Science Foundation Graduate Research Fellowship (to AM), an ARCS Fellowship (to AM), NIH R01 GM080279 (to MG), a James McDonnell Postdoctoral Fellowship (to HS), EMBL core funding and a DFG grant (TY 116/2-1) for SPP1617 (to AT), NIH Director's New Innovator Award DP2OD006466 (to KCH), NSF CAREER Award MCB-1149328 and Awards EF-2125383 and IOS-2032985 (to KCH), and the Allen Discovery Center at Stanford on Systems Modeling of Infection (to KCH). KCH is a Chan Zuckerberg Biohub Investigator. This work was also supported in part by the National Science Foundation under Grant PHYS-1066293 and the hospitality of the Aspen Center for Physics.

## Author contributions

**Matylda Zietek**: Conceptualization; Resources; Data curation; Formal analysis; Validation; Investigation; Visualization; Methodology; Writing—original draft; Writing—review and editing. **Amanda Miguel**: Conceptualization; Resources; Data curation; Software; Formal analysis; Validation; Investigation; Visualization; Methodology; Writing—original draft; Writing—review and editing. **Handuo Shi**: Resources; Data curation; Software; Formal analysis; Validation; Investigation; Visualization; Methodology; Writing—original draft; Writing—review and editing. **Iskander Khusainov**: Resources; Data curation; Software; Formal analysis; Validation; Investigation; Visualization; Methodology. **Abir T Asmar**: Resources; Data curation; Validation; Investigation; Methodology. **Sri Ram**: Resources; Data curation; Formal analysis; Investigation; Visualization; Methodology. **Morgane Wartel**: Validation; Investigation; Methodology. **Anna Sueki**: Resources; Data curation; Formal analysis; Validation; Investigation; Visualization; Methodology. **Martin Schorb**: Investigation; Methodology. **Mark Goulian**: Resources; Supervision; Funding acquisition. **Jean-François Collet**: Supervision. **Martin Beck**: Supervision. **Kerwyn Casey Huang**: Conceptualization; Resources; Data curation; Formal analysis; Supervision; Funding acquisition; Investigation; Visualization; Methodology; Writing—original draft; Project administration; Writing—review and editing. **Athanasios Typas**: Conceptualization; Resources; Data curation; Formal analysis; Supervision; Funding acquisition; Validation; Investigation; Methodology; Writing—original draft; Project administration; Writing—review and editing.

Source data underlying figure panels in this paper may have individual authorship assigned. Where available, figure panel/source data authorship is listed in the following database record: biostudies:S-SCDT-10_1038-S44318-025-00534-w.

## Funding

## Disclosure and competing interests statement

The authors declare no competing interests.

# Expanded View Figures

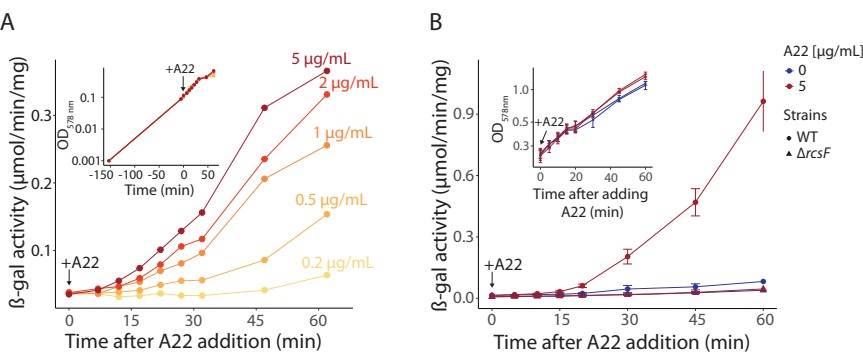

**Figure EV1. Activation of the Rcs system during A22 treatment is dose dependent.**

(A) The Rcs system is activated faster at higher A22 concentration. Activation was measured by monitoring induction of chromosomal *rprA::lacZ*. Cells were treated with A22 at $OD_{578 \, nm} = 0.3$. Inset: growth was unaffected by A22 addition. (B) The Rcs system was activated in wild-type but not Δ*rcsF* cells. Data are $n = 4$ biological replicates of the experiment in Fig. 1A. Data points are mean±1 SD. Source data are available online for this figure.

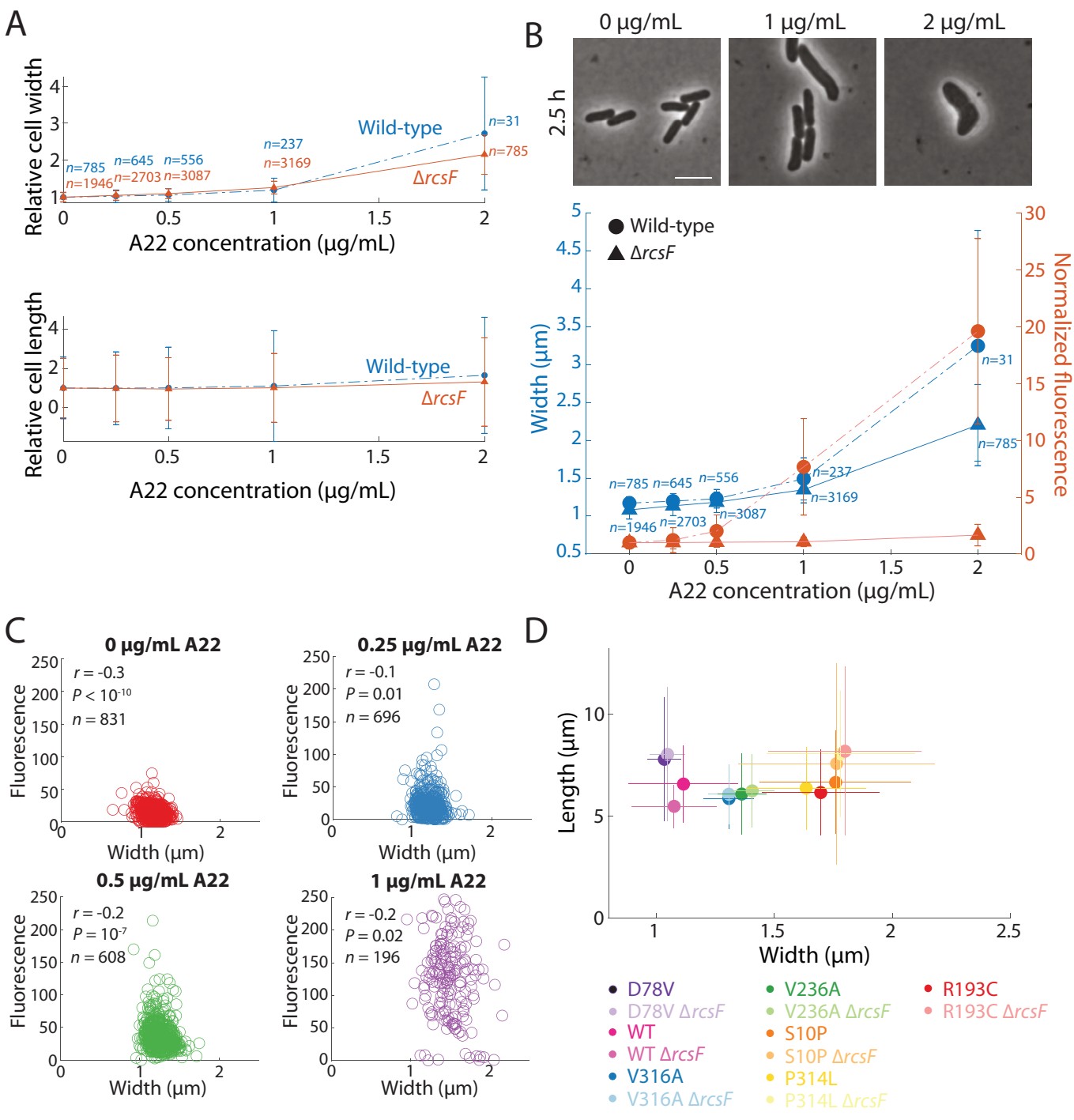

**Figure EV2.** *rprA*-msfGFP expression is dependent on Rcs activation and is not strongly correlated with natural width variation across a population.

(A) Dimensions of A22-treated wild-type and ΔrcsF cells after 2.5 h of A22 treatment, measured relative to the control without treatment. A22-induced changes were not dependent on *rcsF*. Note that *rcsF* cells were more sensitive to A22, and started dying at the A22 concentration 2 μg/mL. For 2 μg/mL A22, wild-type cells increased in width and length more than ΔrcsF cells (P < 0.001 for both cases, two-tailed Student's t tests). The shape of wild-type cells at this concentration was severely deformed from rods and hence had to be excluded from analyses, as length and width could not be defined. ΔrcsF cells died before their shape was severely deformed. Data points are mean ± 1 SD, and n denotes the number of cells analyzed for each condition. (B) Top: images of wild-type cells after 2.5 h of treatment with 0, 1, or 2 μg/mL A22. Scale bar: 5 μm. Bottom: msfGFP intensity from an *rprA* reporter on plasmid pMZ13, as measured via single-cell imaging after 2.5 h of A22 treatment, increased with increasing A22 concentration in wild-type but not ΔrcsF cells. For non-zero A22 concentrations, $P = 10^{-10}$ for fluorescence intensity between WT and ΔrcsF strains, two-tailed Student's t tests. n denotes number of cells analyzed for each condition – dying (ΔrcsF cells) and especially, severe cell deformation (WT) decreased the number of cells analyzed at the highest A22 concentration. Data points are mean±1 SD. (C) The natural variation in cell width across each population after 2.5 h of A22 treatment was not strongly correlated with msfGFP intensity from the *rprA* promoter on pMZ13. r: Pearson's correlation coefficient. P values are from two-tailed Student's t tests. (D) No obvious correlations were observed between the cell width and length of MreB mutants. Data points are mean± 1 SD, with n > 73 cells for each strain. Source data are available online for this figure.

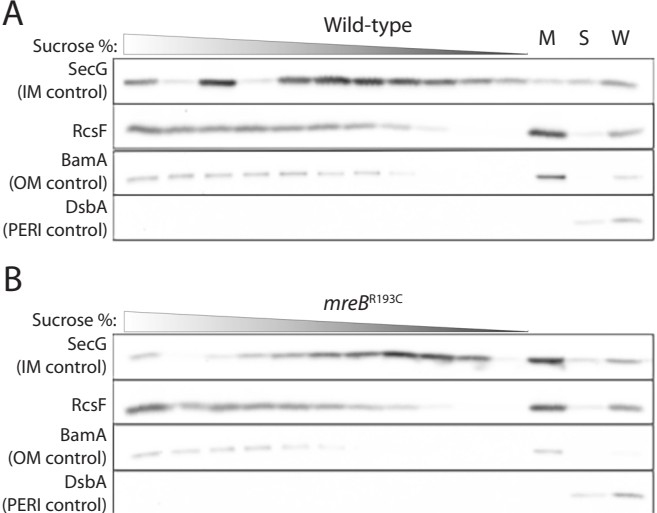

**Figure EV3. RcsF is localized to the outer membrane.**

Sucrose-gradient fractionation shows that RcsF is localized to the outer membrane in both wild-type (**A**) and *mreB*^R193C (**B**) cells. Immunoblotting after sucrose-gradient fractionation of RcsF and other control proteins (SecG for the inner membrane, BamA for the outer membrane, and DsbA for the periplasm). OM: outer membrane, IM: inner membrane, PERI: periplasm, M: total membrane sample prior to fractionation, S: soluble non-membrane fraction, W: whole-cell lysate.

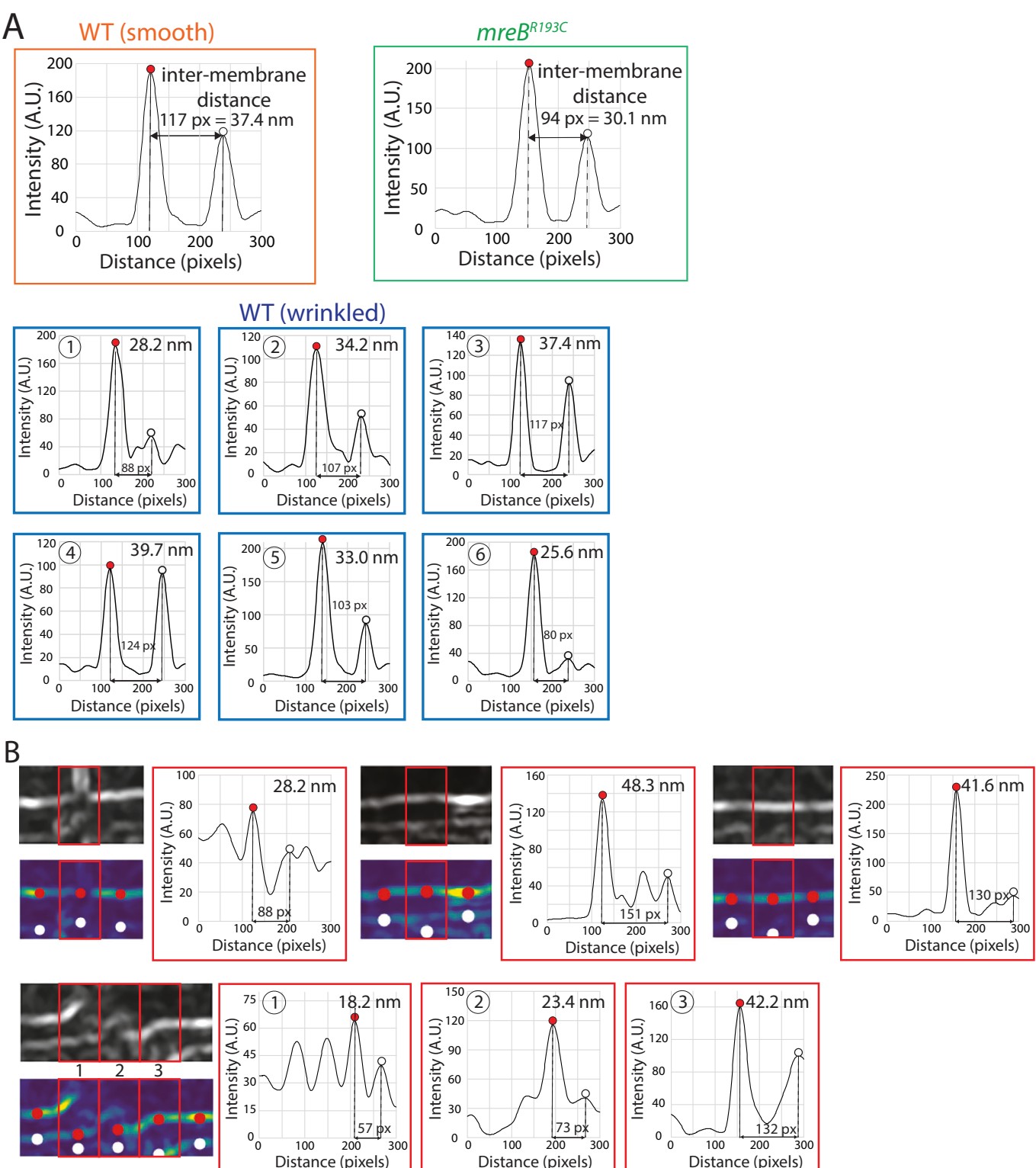

**Figure EV4. Intermembrane distance is calculated based on the distance between bright cross-sections identified as membranes in processed EM images.**

(A) Peaks were identified in electron microscopy image intensity averaged across a 50-nm segment in the direction perpendicular to the membranes. The two highest peaks correspond to the middle of the outer membrane (left peak, red dot) and inner membrane (right peak, white dot). Shown are examples corresponding to the boxes in Fig. 4D (wild-type straight membrane, orange box; wild-type wrinkled membrane, cyan boxes 1–6; $mreB^{R193C}$ straight membrane, green box). px, pixel. (B) Examples of measurements visually identified as erroneous and hence removed due to multiple reasons. Top, from left to right: membrane was discontinuous, a high-contrast object was located in one membrane, insufficient contrast in one membrane. Bottom: incorrect blending of two cryo-EM fields of view. px, pixel.

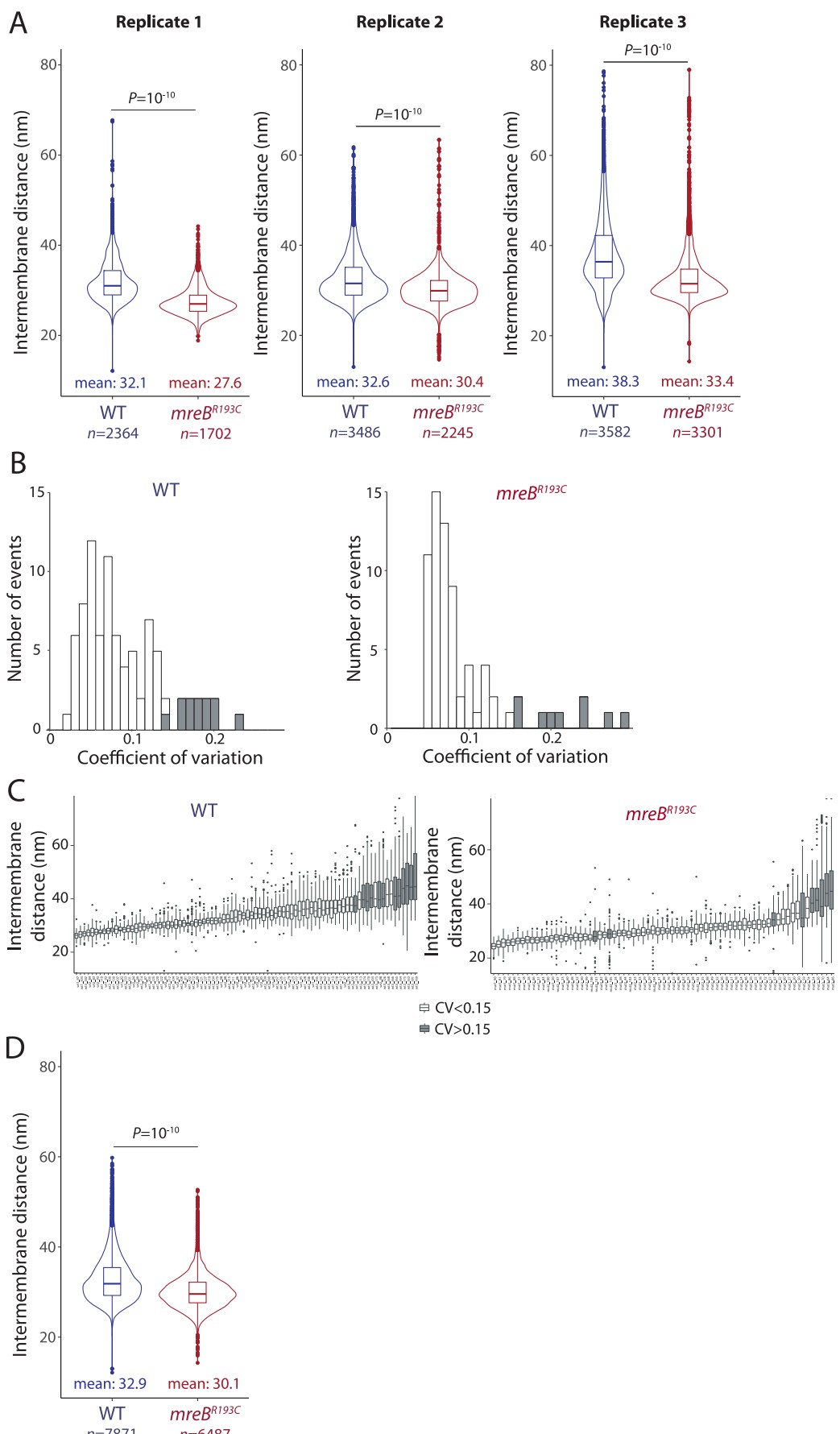

◀ **Figure EV5. The difference in intermembrane distance between wild-type and *mreB^{R193C}* cells is consistent, even though intermembrane distance can vary within a cell, between cells in a population, and between biological replicates.**

(A) Intermembrane distance was consistently higher in wild-type cells compared with *mreB^{R193C}* cells across three biological replicates derived from the same experiment as in Fig. 4B. Means are in nm. *n* represents the number of points at which measurements were made, for ~30 cells of each strain in each replicate. *P* values shown are calculated from two-tailed Student's *t* tests. (B) The distribution of coefficients of variation (CV) across wild-type (left) and *mreB^{R193C}* (right) cells. Cells with CV > 15% are colored in gray. All three biological replicates were included. (C) Intermembrane distances within each wild-type (left) and *mreB^{R193C}* (right) cell, sorted by mean intermembrane distance. Cells with CV > 15% are colored in gray. (D) Intermembrane distance was consistently higher in wild-type cells compared with *mreB^{R193C}* cells when considering only cells with CV < 15% in the dataset presented in Fig. 4B. *n* represents the number of points at which measurements were made, for 74 wild-type and 62 *mreB^{R193C}* cells from three biological replicates pooled together, the *P* value shown was calculated from two-tailed Student's *t* test. Violin plots in (A, D) show the smoothened density of measurements. Box plots in (A, C, D) show the median and first and third quartiles, and the whiskers are 1.5 interquartile ranges from the corresponding quartiles. Data points outside the whiskers are plotted as individual points. Source data are available online for this figure.

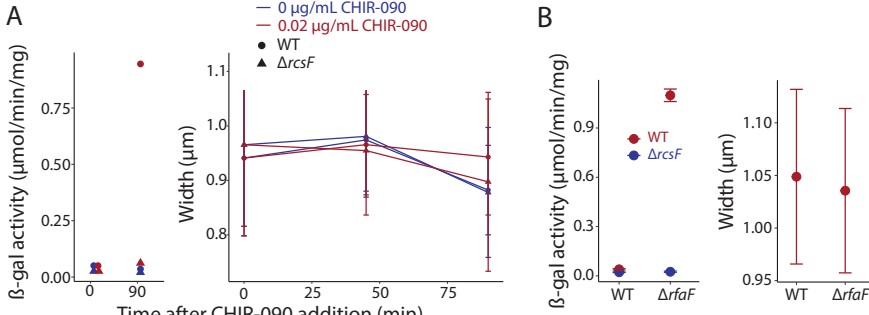

**Figure EV6. Cell width is not affected by CHIR-090 treatment or deletion of *rfaF*.**

(A) Treatment with 0.02 µg/mL CHIR-090 activates the Rcs pathway (left, $n = 1$ replicate) even though cell width is unaffected (right, $n > 200$ cells for each data point). (B) Left: the Rcs pathway is activated in Δ*rfaF* cells. Δ*rfaF* cells had significantly higher Rcs pathway activation compared to wild-type (WT) or Δ*rcsF* cells ($P = 10^{-5}$, $10^{-6}$, and $10^{-6}$, compared to WT, Δ*rcsF*, and Δ*rfaF* Δ*rcsF*, $n = 3$ replicates, two-tailed Student's *t* tests). Right: cell width is largely unaffected by *rfaF* deletion ($n = 1142$ or 926 cells for WT and Δ*rfaF*, respectively). For width measurements in (A, B), data are mean values and error bars represent 1 SD. Source data are available online for this figure.

