## [Peer Review File · The EMBO Journal]

Bacterial cell widening alters periplasmic size and activates envelope stress responses

Kerwyn Huang, Matylda Zietek, Amanda Miguel, Handuo Shi, Iskander Khusainov, Abir Asmar, Sri Ram, Morgane Wartel, Anna Sueki, Martin Schorb, Mark Goulian, Jean-François Collet, Martin Beck, and Athanasios Typas

Corresponding authors: Kerwyn Huang (kchuang@stanford.edu) , Athanasios Typas (typas@embl.de)

Review Timeline:

Submission Date:	1st Sep 22
Editorial Decision:	6th Oct 22
Revision Received:	17th Dec 24
Editorial Decision:	28th Feb 25
Appeal:	18th Mar 25
Editorial Decision:	21st Mar 25
Revision Received:	27th Jun 25
Accepted:	25th Jul 25

Editor: Ieva Gailite

Transaction Report:

Dear Dr. Huang,

Thank you for submitting your manuscript for consideration by the EMBO Journal. We have now received comments from three reviewers, which are included below for your information.

As you will see from the reports, the reviewers find the study interesting, but also indicate several aspects that would have to be strengthened before they can recommend publication here. Based on these positive assessment, I would like to invite a revised version of the manuscript. I think it would be helpful to discuss the revision and the feasibility of particular experiments in more detail via email or phone/videoconferencing - please let me know which option you prefer. I should also add that it is The EMBO Journal policy to allow only a single major round of revision and that it is therefore important to resolve the main concerns at this stage.

We generally allow three months as standard revision time. As a matter of policy, competing manuscripts published during this period will not negatively impact on our assessment of the conceptual advance presented by your study. However, please contact me as soon as possible upon publication of any related work to discuss the appropriate course of action. Should you foresee a problem in meeting this three-month deadline, please contact us to arrange an extension.

When preparing your letter of response to the referees' comments, please bear in mind that this will form part of the Review Process File and will therefore be available online to the community. For more details on our Transparent Editorial Process, please visit our website: <https://www.embopress.org/page/journal/14602075/authorguide#transparentprocess>. Please also see the attached instructions for further guidelines on preparation of the revised manuscript.

Please feel free to contact me if you have any further questions regarding the revision. Thank you for the opportunity to consider your work for publication, and I look forward to discussing your revision.

With best regards,

Ieva

At EMBO Press we ask authors to provide source data for the main and EV figures. Our source data coordinator will contact you to discuss which figure panels we would need source data for and will also provide you with helpful tips on how to upload and organize the files.

We realize that it is difficult to revise to a specific deadline. In the interest of protecting the conceptual advance provided by the work, we recommend a revision within 3 months (4th Jan 2023). Please discuss the revision progress ahead of this time with the editor if you require more time to complete the revisions.

Referee #1:

A common mechanism for bacteria to sense stress is through two-component systems. The Rcs system is used to sense damage to the cell envelope. Antibiotics that target the elongation machinery are more effective in cells mutant for the Rcs system. It is currently unclear why this is. Because these drugs also effect cell size, the authors examine whether the Rcs system is able to sense and respond to changes in cell size. Interestingly, it was found that changes in cell width lead to changes in the width of the periplasm and the authors show evidence that RcsF activity changes based on the size of the periplasm.

While overall I believe the experiments are well thought out and address and important biological questions there are some points that if addressed would help to strengthen the manuscript. First, the authors discuss the mean width of different strains multiple times but never show statistical significance. Second, while the authors show three different ways to change cell shape (drugs, mutations, mechanical stress), they often follow up future experiments only testing one of these and then make a broad conclusion (see point 4-5 below) for cell shape perturbations.

Major Issues:

- 1) The authors comment multiple times about means being different yet do not show any P-value or other statistical test to confirm this.
- 2) line 137: The authors state that A22-induced cell width changes are Rcs-independent but this is not shown in Fig.1B. An rcsF mutant would need to be included to make this claim.
- 3) Fig 1E. Why does the activation of the Rcs dip after 30 min when cells are treated with 5 ug/ml of A22?
- 4) Mechanical sensing: The authors state that they cells of varying deformations and only see fluorescence in the fatter cells (line 550-551). However, in the second image in Fig. 3B it appears that all the cells in the bottom left cluster have signal even though some are thinner than the middle cluster which shows no signal. This is the same as the upper right cluster which clearly has large cells yet no signal. It is interesting that there is signal only on the left side suggesting a possible error with the illumination field?

To address the idea of mechanical perturbations activating the Rcs I think it would be cleaner if the authors would grow cells in channels of different widths. The authors would be able to make both skinnier and fatter cells in a controlled manner which help define a width range needed to activate the system. In addition, the authors could correlate the periplasmic space in these cells to further support their hypothesis about periplasmic size being important for RcsF activation.

5) Periplasmic width: It would be nice to see this done in another condition (see above) to show that it is a general mechanism for shape mutants to activate Rcs. Do cells treated with A22 have a different periplasmic size?

The authors should also put their +7 Rcs mutant into the cells with the MreB point mutations/A22 treated cells. If the change in periplasmic width is responsible than the +7 version should have no increase.

Minor Issue:

line 138-139: Grinnel et al (PMID 35115684) would be a good reference for A22 not slowing down cell growth.

Fig. 2B: Please include WT

Fig S2A: are the images from WT or rcsF cells? It would be useful to see both; The Y axis should read fluorescence (it is missing the fl)

Referee #2:

General Comments: In this manuscript the authors explore the effects of cell widening on periplasmic cell size and its effect on activation of the RcsF signaling system. The data is of high quality and the manuscript is written in a thoughtful and informative manner. The authors find that cell widening is associated with activation of the RcsF system and this most likely a result of physical narrowing of the periplasm with subsequent result that the RcsF lipoprotein is in closer proximity to the inner membrane sensor for the signal transduction system. Important experiments include lengthening of RcsF through a protein linker with constitutive activation as well as measurement of the periplasmic size in cells with mutations that alter cell width. The manuscript is well done and some minor issues should be addressed prior to publication.

Specific comments:

Figure 2C. It would be nice to see some statistical analysis of the data, though it certainly looks significant, this should be done prior to publication since this is a critical piece of data for the manuscript.

The RcsF+7 data is convincing as is the protein localization, it would be nice to show by expression induction if possible that the proteins are of equal stability and that the blotting was done such that a small fragment of RcsF has not been generated that activates the machinery, and are a discussion of this less likely possibility.

Discussion: The authors may want to be cautious about their statement that defects in LPS and the Bam machinery do not alter periplasmic thickness. Defects in LPS and the Bam machinery may also alter periplasmic thickness at a subset of sites across the membrane or they may simply alter the localization of RcsF, but they are non-physiologic conditions. Conditions of membrane damage such as cationic antimicrobial peptides may alter the periplasmic space at places of damage and bring RcsF closer to the inner membrane. Therefore, alteration in the periplasmic space may be necessary and sufficient for activation of RcsF. Such conditions may include membrane damage, cellular defects in mutants as well as cell division.

Referee #3:

The Rcs phosphorelay in enterobacteria plays an important role in the cellular response to cell envelope stress. One puzzling aspect of this system is that multiple stress signals, with what seem like very different targets, all induce the Rcs response, via an outer membrane lipoprotein, RcsF. RcsF in turn interacts with a negative regulator of the system, IgaA. The current model for induction suggests that RcsF, upon activation, interacts with IgaA, releasing IgaA negative regulation of the phosphorelay. However, the range of inducing signals makes it difficult to understand what is being sensed by RcsF. These inducers include A22, targeting MreB, osmotic shock, antimicrobials such as polymyxin, targeting OM phospholipids and LPS as well as number of mutants. The manuscript by Zietek et al. uses sophisticated imaging analysis to provide evidence that many of the inducing conditions lead to changes in cell shape (cell widening), and that this cell shape change results in a thinner periplasm. This leads the authors to propose that the decreased intermembrane distance brings RcsF, extending into the periplasm from the outer membrane, closer to inner membrane localized IgaA, thereby allowing the necessary interaction for Rcs activation. An RcsF mutant (RcsF+7) with a longer linker region also activates the system, consistent with this model. The idea that periplasmic size is critical for RcsF signaling was previously suggested by work by Collet and colleagues (Asmar et al, 2017), in which lpp mutants increased the periplasmic size and blocked signalling. Here, the authors show a strong correlation in the other direction - decreasing periplasmic size is associated with induction. That said, the majority of the data is correlative.

Points requiring attention:

1. There is a 15' delay in both A22 signaling and in the apparent width change (Fig. 1A, B). What is happening during that 15'? Is there induction of any other cell changes, or is this the time necessary for the growing cells to show a measurable change in width?
2. Given the correlation of cell width and Rcs induction, it is important to understand whether the cell shape change induces Rcs (suggested here), or inducing Rcs leads to a cell shape change. Lines 136-137 say that Fig. 1B shows that A22-induced changes in cell size are Rcs independent, but this is not in the figure. Is that relevant data that in Fig. S2A? Maybe this should be in a main figure (with error bars that can be better assigned to each line), possibly also with an rcsB mutant, to fully block Rcs activity.
3. Fig. S2A: There appears to be some difference between WT and del rcsF width here. In Fig. 2B, the width of the R193C mutant appears to be significantly less than for the R193C del rcsF double mutant (in the opposite direction to that seen with A22). Do these suggest that there is an effect of Rcs on cell width?
4. In Fig. 1B, would the data look the same for cells devoid of rcsF? Can that plot be provided for comparison (that would also help to make the point about the cell width effect being independent of rcsF, if true). This would provide earlier time points than Fig. S2A if there is a more transient effect of Rcs, as well.
5. Is it possible to estimate, from the data in Fig. 2, how small a difference in cell width is sufficient to induce Rcs?
6. The focus here is on cell width, with length only plotted for Fig. 1. Would length show changes in the mutants, etc.? Does this

correlate with Rcs, or only width does?

7. Previously, it has been shown that overexpression of RcsF+7 leads to a higher PrprA-lacZ activity in a lpp+14 mutant as compared to pRcsF (both in the presence and absence of A22) and that it helps recover the Rcs signaling in an Rcs-impaired mdoG lpp+14 strain. The current study also uses this RcsF+7 mutant to show that its overexpression induces Rcs in a WT background as well. Are these solely due to the extension in RcsF length due to the added residues linker region or are there additional conformational changes involved (such as changes in interaction with Bam/OMPs/IgaA)? See the next, related point.
8. As noted by the authors, not all inducing conditions appear to affect periplasmic size. The data fits a model in which one necessary component of RcsF-dependent signaling is that RcsF be able to (better) reach IgaA, and thinning the periplasm may be sufficient to achieve this. How do the authors then think about induction by LPS perturbing treatments? Is that qualitatively different? Would it be affected, positively or negatively, by thinning or thickening the periplasm?
9. Minor point: The typo in the Y-axis of Fig. S2A (fluorescence) needs to be edited.

Referee #1:

A common mechanism for bacteria to sense stress is through two-component systems. The Rcs system is used to sense damage to the cell envelope. Antibiotics that target the elongation machinery are more effective in cells mutant for the Rcs system. It is currently unclear why this is. Because these drugs also effect cell size, the authors examine whether the Rcs system is able to sense and respond to changes in cell size. Interestingly, it was found that changes in cell width lead to changes in the width of the periplasm and the authors show evidence that RcsF activity changes based on the size of the periplasm.

While overall I believe the experiments are well thought out and address and important biological questions there are some points that if addressed would help to strengthen the manuscript. First, the authors discuss the mean width of different strains multiple times but never show statistical significance. Second, while the authors show three different ways to change cell shape (drugs, mutations, mechanical stress), they often follow up future experiments only testing one of these and then make a broad conclusion (see point 4-5 below) for cell shape perturbations.

We appreciate the reviewer's positive comments! We have addressed all of their comments and concerns in detail below.

Major Issues:

1) The authors comment multiple times about means being different yet do not show any P-value or other statistical test to confirm this.

We have now ensured that statistical tests are included with all comparisons.

2) line 137: The authors state that A22-induced cell width changes are Rcs-independent but this is not shown in Fig. 1B. An rcsF mutant would need to be included to make this claim.

These data are included in Fig. S2. We have now clarified this reference in the text.

3) Fig 1E. Why does the activation of the Rcs dip after 30 min when cells are treated with 5 ug/ml of A22?

There are several reasons for this decrease that we can postulate. Once cells become sufficiently enlarged, it may be that the mechanism of Rcs activation is compromised. Additionally, because the data in Fig. 1E are the rate of fluorescence increase, which is a function of both Rcs activation and the dynamics of fluorescent protein folding and maturation, there may also be delays from these other processes. Moreover, in a previous study (Silvis *mBio* 2021), we showed that increases in cell width (including due to A22 treatment) lead to increased expression of *mreB*, which may contribute partially to the decrease. We note that the rate of beta-gal increase in Fig. 1A (single-copy transcriptional fusion) is relatively stable, suggesting that fluorescent protein

folding/maturation or promoter saturation may be the causes of the dip in activation in Fig. 1E. Regardless, we have now switched this panel to report on the fluorescence itself, which we felt was more clear for the reader.

4) Mechanical sensing: The authors state that they cells of varying deformations and only see fluorescence in the fatter cells (line 550-551). However, in the second image in Fig. 3B it appears that all the cells in the bottom left cluster have signal even though some are thinner than the middle cluster which shows no signal. This is the same as the upper right cluster which clearly has large cells yet no signal. It is interesting that there is signal only on the left side suggesting a possible error with the illumination field?

This difference is likely due to the high induction of the parent cells that generated the bottom left cluster, such that the daughter cells were generated in a thinner state but retained high GFP signal due to activation signal from the parent cells. We have modified the text to acknowledge this variation, and to focus on the correlation shown in Fig. 3C.

To address the idea of mechanical perturbations activating the Rcs I think it would be cleaner if the authors would grow cells in channels of different widths. The authors would be able to make both skinnier and fatter cells in a controlled manner which help define a width range needed to activate the system. In addition, the authors could correlate the periplasmic space in these cells to further support their hypothesis about periplasmic size being important for RcsF activation.

The reviewer brings up an interesting idea, but we are not aware of any strategy for altering cell shape through growth in channels except in the context of L-forms that lack cell walls (such as the work of Cees Dekker) and hence have altered physiology. Note that even with such channels, it would be challenging to extract the cells for CryoEM. In the absence of such a strategy, we elected to focus on relatively controlled perturbations involving the MreB mutants and compression.

5) Periplasmic width: It would be nice to see this done in another condition (see above) to show that it is a general mechanism for shape mutants to activate Rcs. Do cells treated with A22 have a different periplasmic size?

We focused on the MreB mutant because cells are not at steady-state under A22 treatment, as evidenced by the dynamics of *rprA* fluorescence in Fig. 1E and our previous finding that cell width changes result in upregulation of MreB levels (Silvis *mBio* 2021). Moreover, in our companion paper by Miguel et al. (which is included along with this revision), we find that stresses such as A22 treatment alter the physiological response to Rcs activation. Thus, while we would also find it interesting to investigate more broadly whether periplasmic width is generally changing in shape mutants, we think that MreB mutation is the cleanest way to open this line of inquiry and that a comprehensive interrogation of the correlation would be extensive and beyond the scope of this study.

The authors should also put their +7 Rcs mutant into the cells with the MreB point mutations/A22 treated cells. If the change in periplasmic width is responsible than the +7 version should have no increase.

The reviewer brings up an interesting scenario, which unfortunately would involve technical difficulties to address. Unfortunately, the plasmids for the MreB mutant and pAM238 with *rcsF*⁺⁷ have the same pSC101 origin, and hence are not compatible, complicating efforts to address this question. Even if we were to go down the laborious line of making new plasmids/constructs (MreB plasmids are in an *mreBCD* mutant background), we are unclear if such experiment would give a clear answer. We observed close to maximum Rcs activation by both *RcsF*⁺⁷ and in the *MreB*^{R193C} mutant (as measured by the *rprA* promoter), so no difference in the combination could stem from epistasis, as the reviewer suggests, but it could also be the case that activation is maxed out.

Minor Issue:

line 138-139: Grinnel et al (PMID 35115684) would be a good reference for A22 not slowing down cell growth.

We have added this citation.

Fig. 2B: Please include WT

We have now included the wild-type strain in this plot along with a Δ *rcsF* strain with *MreB*^{WT}, whose behavior was consistent with the trend for the *MreB* mutants.

Fig S2A: are the images from WT or rcsF cells? It would be useful to see both; The Y axis should read fluorescence (it is missing the fl)

The images are from WT cells, which is now noted in the legend. Fluorescence is labeled on the right-hand side of the plot in orange.

Referee #2:

General Comments: In this manuscript the authors explore the effects of cell widening on periplasmic cell size and its effect on activation of the RcsF signaling system. The data is of high quality and the manuscript is written in a thoughtful and informative manner. The authors find that cell widening is associated with activation of the RcsF system and this most likely a result of physical narrowing of the periplasm with subsequent result that the RcsF lipoprotein is in closer proximity to the inner membrane sensor for the signal transduction system. Important experiments include lengthening of RcsF through a protein linker with constitutive activation as well as measurement of the periplasmic size in cells with mutations that alter cell width. The manuscript is well done and some minor issues should be addressed prior to publication.

We appreciate the reviewer's positive comments! We have addressed their comments and questions in detail below.

Specific comments:

Figure 2C. It would be nice to see some statistical analysis of the data, though it certainly looks significant, this should be done prior to publication since this is a critical piece of data for the manuscript.

We have now included statistical tests for all comparisons.

The RcsF+7 data is convincing as is the protein localization, it would be nice to show by expression induction if possible that the proteins are of equal stability and that the blotting was done such that a small fragment of RcsF has not been generated that activates the machinery, and are a discussion of this less likely possibility.

The Western blot below shows that RcsF⁺⁷ and RcsF^{WT} are expressed at similar levels. The white boxes highlight two biological replicates of RcsF in strain AA73 ($\Delta rcsF$ pSC202 *rcsF*^{WT}), while the gray boxes highlight two biological replicates of RcsF in strain AA348 ($\Delta rcsF$ pSC202 *rcsF*⁺⁷).

Discussion: The authors may want to be cautious about their statement that defects in LPS and the Bam machinery do not alter periplasmic thickness. Defects in LPS and the Bam machinery may also alter periplasmic thickness at a subset of sites across the membrane or they may simply alter the localization of RcsF, but they are non-physiologic conditions. Conditions of membrane damage such as cationic antimicrobial peptides may alter the periplasmic space at places of damage and bring RcsF closer to the inner membrane. Therefore, alteration in the periplasmic space may be necessary and sufficient for activation of RcsF. Such conditions may include membrane damage, cellular defects in mutants as well as cell division.

The reviewer brings up excellent points. Our point was to emphasize the distinction with certain perturbations that do not show a change in cell width, but we agree that there are many ways in which the periplasmic space could be altered. We have rewritten the text to be more cautious about our claims about defects in LPS and the BAM machinery. Certain conditions are challenging to address with CryoEM, such as

polymyxin treatment, due to the transient nature of the perturbation that is quickly addressed by cells.

Referee #3:

The Rcs phosphorelay in enterobacteria plays an important role in the cellular response to cell envelope stress. One puzzling aspect of this system is that multiple stress signals, with what seem like very different targets, all induce the Rcs response, via an outer membrane lipoprotein, RcsF. RcsF in turn interacts with a negative regulator of the system, IgaA. The current model for induction suggests that RcsF, upon activation, interacts with IgaA, releasing IgaA negative regulation of the phosphorelay. However, the range of inducing signals makes it difficult to understand what is being sensed by RcsF. These inducers include A22, targeting MreB, osmotic shock, antimicrobials such as polymyxin, targeting OM phospholipids and LPS as well as number of mutants. The manuscript by Zietek et al. uses sophisticated imaging analysis to provide evidence that many of the inducing conditions lead to changes in cell shape (cell widening), and that this cell shape change results in a thinner periplasm. This leads the authors to propose that the decreased intermembrane distance brings RcsF, extending into the periplasm from the outer membrane, closer to inner membrane localized IgaA, thereby allowing the necessary interaction for Rcs activation. An RcsF mutant (RcsF+7) with a longer linker region also activates the system, consistent with this model. The idea that periplasmic size is critical for RcsF signaling was previously suggested by work by Collet and colleagues (Asmar et al, 2017), in which lpp mutants increased the periplasmic size and blocked signalling. Here, the authors show a strong correlation in the other direction - decreasing periplasmic size is associated with induction. That said, the majority of the data is correlative.

Points requiring attention:

1. There is a 15' delay in both A22 signaling and in the apparent width change (Fig. 1A, B). What is happening during that 15'? Is there induction of any other cell changes, or is this the time necessary for the growing cells to show a measurable change in width?

The reviewer brings up a good point – the short answer is that we do not know the origin of the 15' delay in A22 signaling and the width change. We showed these data to emphasize that width and Rcs activation are correlated. There is no delay in Rcs activation in other growth conditions. The delay might be growth rate or growth phase dependent, but since this point is tangential to the subject of the paper, we have not delved into it in greater detail.

2. Given the correlation of cell width and Rcs induction, it is important to understand whether the cell shape change induces Rcs (suggested here), or inducing Rcs leads to a cell shape change. Lines 136-137 say that Fig. 1B shows that A22-induced changes in cell size are Rcs independent, but this is not in the figure. Is that relevant data that in Fig. S2A? Maybe this should be in a main figure (with error bars that can be better assigned to each line), possibly also with an rcsB mutant, to fully block Rcs activity.

We apologize, Fig. S2A is the relevant data for this point; we have modified the text accordingly.

3. Fig. S2A: There appears to be some difference between WT and $\Delta rcsF$ width here. In Fig. 2B, the width of the R193C mutant appears to be significantly less than for the R193C $\Delta rcsF$ double mutant (in the opposite direction to that seen with A22). Do these suggest that there is an effect of Rcs on cell width?

While there was a small difference in cell width between WT and $\Delta rcsF$ in Fig. S2A, these differences were much smaller than the increases in WT cells due to A22 treatment. Moreover, it is important to note that $\Delta rcsF$ mutants are more sensitive to A22 (we have noted this now in the Figure legend). The width difference between R193C cells with and without *rscF* is difficult to interpret because of distortions to the rod-shaped morphology.

*4. In Fig. 1B, would the data look the same for cells devoid of *rscF*? Can that plot be provided for comparison (that would also help to make the point about the cell width effect being independent of *rscF*, if true). This would provide earlier time points than Fig. S2A if there is a more transient effect of Rcs, as well.*

For $\Delta rcsF$ data as a function of time, we refer the reader to our companion paper, which is now included with this revision.

5. Is it possible to estimate, from the data in Fig. 2, how small a difference in cell width is sufficient to induce Rcs?

Our data from the MreB mutants in Fig. 2A,B suggest that Rcs activation is not due to width passing a particular threshold, but rather tuning such that any changes in cell width (including decreases) will be linearly correlated with activation level. The addition of wild-type to Fig. 2B helps to emphasize this point.

6. The focus here is on cell width, with length only plotted for Fig. 1. Would length show changes in the mutants, etc.? Does this correlate with Rcs, or only width does?

We have included length measurements in Fig. 1B, which show that length does not correlate with Rcs activation to the same degree as width. However, length and width are to coupled to some degree.

In the MreB mutants, length and width are not correlated, as we now show in Fig. EV2D.

*7. Previously, it has been shown that overexpression of *RcsF+7* leads to a higher *PrprA-lacZ* activity in a *lpp+14* mutant as compared to *pRcsF* (both in the presence and absence of A22) and that it helps recovers the Rcs signaling in an Rcs-impaired *mdoG lpp+14* strain. The current study also uses this *RcsF+7* mutant to show that its*

overexpression induces Rcs in a WT background as well. Are these solely due to the extension in RcsF length due to the added residues linker region or are there additional conformational changes involved (such as changes in interaction with Bam/OMPs/IgaA)? See the next, related point.

RcsF⁺⁷ responds to A22 and mecillinam like RcsF^{WT}, suggesting that interaction with the BAM complex and OMPs is similar between the two proteins.

8. As noted by the authors, not all inducing conditions appear to affect periplasmic size. The data fits a model in which one necessary component of RcsF-dependent signaling is that RcsF be able to (better) reach IgaA, and thinning the periplasm may be sufficient to achieve this. How to the authors then think about induction by LPS perturbing treatments? Is that qualitatively different? Would it be affected, positively or negatively, by thinning or thickening the periplasm?

There are certainly Rcs-activating conditions that are qualitatively different, such as Δrfa mutants that do not alter cell size, and LPS perturbations such as CHIR-090 treatment falls into this scenario (Fig. S6). Polymyxin stress causes only transient Rcs activation, and damage is addressed quickly. Our model would predict that in the Δrfa mutant or during transient polymyxin activation, alteration of periplasmic size could explain the much higher Rcs activation relative to wild type (Fig. S6B). We have added speculation about these points to the Discussion.

9. Minor point: The typo in the Y-axis of Fig. S2A (fluorescence) needs to be edited.

We have ensured that this label is correct.

Dear KC and Nassos,

Thank you for submitting your revised manuscript for consideration by The EMBO Journal. We have now received comments from two of the original reviewers, which are included below for your information.

As you can see, while reviewer #3 is broadly satisfied with the revision, reviewer #1 maintains that their initial concerns have not been satisfactorily addressed and finds that further experimental analysis would be needed to support the proposed role of periplasmic size in RcsF activation. Since our policy supports a single major revision round, and it remains unclear whether the outstanding questions can be further addressed experimentally, I am afraid that we cannot offer further steps towards publication in The EMBO Journal.

In the interest of rapid publication of the study, I have discussed your manuscript and the referee reports with my colleague Achim Breiling at our sister journal EMBO Reports. I am happy to say that Achim finds the work relevant for EMBO Reports, and would be interested to proceed with the manuscript.

EMBO reports emphasizes novel functional over detailed mechanistic insight and therefore will not require that points regarding more refined mechanistic details are addressed. However, all remaining concerns regarding the major conclusions of the study indicated by referee #1 and all technical and experimental limitations mentioned and points regarding data presentation need to be addressed in a further revised manuscript.

Please feel free to contact Achim directly (a.breiling@emboreports.org), if you have any questions regarding the revision. If you are interested in this option, please use the transfer link below.

Link Not Available

Thank you in any case for the opportunity to consider this manuscript. I am sorry that I could not communicate more positive news, and I sincerely hope that you will find the transfer option of interest.

With kind regards,

leva

leva Gailite, PhD
Senior Scientific Editor
The EMBO Journal
Meyerhofstrasse 1
D-69117 Heidelberg
Tel: +4962218891309
i.gailite@embojournal.org

Referee #1:

A common mechanism for bacteria to sense stress is through two-component systems. The Rcs system is used to sense damage to the cell envelope. Antibiotics that target the elongation machinery are more effective in cells mutant for the Rcs system. It is currently unclear why this is. Because these drugs also effect cell size, the authors examine whether the Rcs system is able to sense and respond to changes in cell size. Interestingly, it was found that changes in cell width lead to changes in the width of the periplasm and the authors show evidence that RcsF activity changes based on the size of the periplasm.

I would like to thank the authors for addressing some of the previous comments, especially toning down the language about their conclusions, they still seem to have missing statistic and in correct figure labels.

Major Issues:

The authors report cell shape changes in different conditions or mutant background but report that the values are based on very low number of cells per condition (30, 45 etc., (EV2, 2 etc.). It is unclear if these experiments were repeated and if the data shown is for a representative experiment or pooled from replicates. If pooled, that would mean we are lucky to have 15 cells per

experiment. This number does not seem acceptable to make conclusions. If these are not pooled then the authors need to state that and inform us about the number of replicates.

Either way I would expect to see many more cells analyzed. The authors did measure an appropriate amount of cells for 1B (if a single experiment) and 2C so it is not clear why other experiments are so under imaged.

Fig. 2B: Is the effect if Rcs activation stronger in wide mutants than thinner. While the authors only test 1 thin mutant (D78V) it appears that there is no change in activation. There does not appear to be a statistical test to indicated if these two strains are indeed different in Rcs activation.

The authors claim that the loss of rcsF has no effect on cell width, but it appears that in mutant R193C the rcsF mutant is much wider (as pointed out in the previous review). Can the authors discuss this. It maybe that not enough cells were imaged as the error bar for width is quite large.

The authors are right that putting rcsF+7 into the fat mutants would be hard to interpret, however the authors should make a rcsF-linker strain that reduces the length of the linker and introduce this into some of the fatter cells. Does this return the cells to WT expression levels now that the distance between rcsF and lgaA is reduced.

Minor Issues:

Line 134: mean cell width increased in an A22 dose-dependent manner (Fig. EV1A). There is no width measurement in EV1A, it is a graph showing Bgal v time in A22.

EV2A: statistical difference between rcsF and WT are needed as the error bars appear to overlap at A22 2 ug/ml. See above comment about cell numbers and replicates.

The authors reference Grinnell et al. which showed that cells will continue to grow in the presence of A22 in the conditions the authors tested in this study (line 136-137). The authors then state that the rcsF cells begin to die in the EV2 legend. Where these grown in the same manner as stated in the materials and methods for fig 1 (line 350). Please include this information for all experiments where A22 is used.

Referee #3:

The revised manuscript provides evidence for a strong positive correlation of cell width with Rcs induction, and a negative correlation with periplasmic thickness. The revision improves the presentation significantly. This is an interesting and potentially important observation, and suggest that components of the Rcs regulon may play roles in cell shape regulation (in part investigated in a second study by this group).

** As a service to authors, EMBO Press provides authors with the possibility to transfer a manuscript that one journal cannot offer to publish to another EMBO publication or the open access journal Life Science Alliance launched in partnership between EMBO Press, Rockefeller University Press and Cold Spring Harbor Laboratory Press. The full manuscript and if applicable, reviewers' reports, are automatically sent to the receiving journal to allow for fast handling and a prompt decision on your manuscript. For more details of this service, and to transfer your manuscript please click on Link Not Available. **

Dear Ieva,

thanks for sending the reviews back. Both KC and I were traveling and needed to chat before replying, hence delays.

We read the reviews, and we believe that all the points that reviewer 1 raises (some of which are new to this revision) can be addressed immediately (see below). They are also minor and do not question any of the major messages of the paper. **Hence, we would appreciate you reconsidering. Happy to talk further in person.**

In terms of numbers of cells (new point), we had listed in Figures that we used ">30/45 cells" when showing multiple mutants to save space. We can certainly specify numbers of cells analysed (come from one experiment to answer reviewer's question), have more cells analysed when needed), and provide the statistic test in the single case missing.

Importantly, in terms of the experiment requested (an RcsF with shorter linker in fat cells), this experiment is impossible to do as shortening the linker of RcsF affects its correct translocation to the outer membrane, as shown before by Jean-Francois's lab (PMID: 34326538).

Thanks for your consideration.

Best wishes,

Nassos & KC

Dear Nassos and KC,

Thank you for contacting me regarding the recent decision on your manuscript and taking time to discuss the study in person. I am glad to hear that you will be able to address the remaining comments by reviewer #1. I therefore invite you to submit a revised version with the additional data included as requested, apart from point 5, for which please provide a textual explanation regarding its feasibility.

Furthermore, please address the following formatting issues:

1. Please check that the funding information is correct and identical both in the manuscript and our online system.
2. CRediT has replaced the traditional author contributions section because it offers a systematic, machine-readable author contributions format that allows for more effective research assessment. Please remove the Authors Contributions from the manuscript and use the free text boxes beneath each contributing author's name in our online submission system to add specific details on the author's contribution. More information is available in our guide to authors.
3. Please remove the main and Expanded View figures from the manuscript text and upload as individual, high resolution figure files. The legends should stay in the manuscript text, after the References section. The heading for the EV figure legends should be changed from "Suppl. Figure legends" to "Expanded View Figure Legends".
4. Please rename the section "Code Availability" into "Data Availability". The second "Data Availability" section can be removed.
5. Figure panel 5B is not mentioned in the manuscript text.
6. Please update references according to The EMBO Journal style - where there are more than 10 authors on a paper, the first 10 should be listed, followed by 'et al.' DOIs should be removed, unless provided for datasets and preprints. Please see further information here: <https://www.embopress.org/page/journal/14602075/authorguide#referencesformat>
7. Please remove the Reagents and Tools Table from the manuscript and upload it as a separate file choosing the file type "Reagent Table" using the template that you can find in our author guidelines:
<https://www.embopress.org/page/journal/14602075/authorguide#structuredmethods>
8. Our data editors have flagged the following issues in figure legends that need correcting:
 - Please provide the exact p values in the legends of figures 1E, 2C, D, 4B, 5A; EV2 B, C, EV5 A, D; EV6 B.
 - Please define the box plots in terms of minima, maxima and percentile in the legends of figures 4B, EV5 A, D.
 - Please define the box plots in terms of minima, maxima, centre, bounds of box and whiskers, and percentile in the legends of figures EV5 C.
 - Please provide information on the number and nature of replicates in the legend of figure EV5 D.
 - Please define the error bars in the legends of figures EV1 B; EV2 B.
 - Please define the measure of center for the error bars in the legends of figures 5A, EV2 A.
7. Please note that the scale bar needs to be defined for figure EV2 B
9. Papers published in The EMBO Journal are accompanied online by a 'Synopsis' to enhance discoverability of the manuscript. It consists of A) a short (1-2 sentences) summary of the findings and their significance, B) 3-4 bullet points highlighting key results and C) a synopsis image that is 550x300-600 pixels large (width x height, jpeg or png format). You can either show a model or key data in the synopsis image. Please note that the image size is rather small and that text needs to be readable at the final size. Please send us this information together with the revised manuscript.

Finally, at EMBO Press we ask authors to provide source data for the main and EV figures. Our source data coordinator Hannah Sonntag will contact you to discuss which figure panels we would need source data for and will also provide you with helpful tips on how to upload and organise the files.

Please feel free to contact me if have any further questions regarding the revision. Thank you for the opportunity to consider your work for publication, and I look forward to receiving the final version fo your manuscript.

With best wishes,

leva

leva Gailite, PhD
Senior Scientific Editor
The EMBO Journal
Meyershofstrasse 1
D-69117 Heidelberg
Tel: +4962218891309
i.gailite@embojournal.org

We realize that it is difficult to revise to a specific deadline. In the interest of protecting the conceptual advance provided by the work, we recommend a revision within 3 months (19th Jun 2025). Please discuss the revision progress ahead of this time with the editor if you require more time to complete the revisions.

Referee #1:

A common mechanism for bacteria to sense stress is through two-component systems. The Rcs system is used to sense damage to the cell envelope. Antibiotics that target the elongation machinery are more effective in cells mutant for the Rcs system. It is currently unclear why this is. Because these drugs also effect cell size, the authors examine whether the Rcs system is able to sense and respond to changes in cell size. Interestingly, it was found that changes in cell width lead to changes in the width of the periplasm and the authors show evidence that RcsF activity changes based on the size of the periplasm.

I would like to thank the authors for addressing some of the previous comments, especially toning down the language about their conclusions, they still seem to have missing statistic and in correct figure labels.

Major Issues:

The authors report cell shape changes in different conditions or mutant background but report that the values are based on very low number of cells per condition (30, 45 etc., (EV2, 2 etc.). It is unclear if these experiments were repeated and if the data shown is for a representative experiment or pooled from replicates. If pooled, that would mean we are lucky to have 15 cells per experiment. This number does not seem acceptable to make conclusions. If these are not pooled then the authors need to state that and inform us about the number of replicates.

Either way I would expect to see many more cells analyzed. The authors did measure an appropriate amount of cells for 1B (if a single experiment) and 2C so it is not clear why other experiments are so under imaged.

We agree with the reviewer that more cells can strengthen our conclusions. For Fig 2B, we have analyzed additional images from the same experiment so that most mutants now contain ~100-300 cells. For Fig EV2A-B, the sole condition with low n ($n=31$) is the highest A22 concentration, which caused substantial cell deformation for wildtype cells and could not be analyzed for width and length as most cells were not rods anymore. All other conditions had several hundreds of cells. In both new figures (Fig 2 and EV2), we have now explicitly listed out the number of cells analyzed for each mutant/condition.

Fig. 2B: Is the effect if Rcs activation stronger in wide mutants than thinner. While the authors only test 1 thin mutant (D78V) it appears that there is no change in activation. There does not appear to be a statistical test to indicated if these two strains are indeed different in Rcs activation.

We have performed a statistical test showing that the D78V mutant indeed has lower Rcs activation than WT ($p < 0.001$, two-tailed Student's t -test). We have incorporated this result in the main text (lines 169-171).

The authors claim that the loss of rcsF has no effect on cell width, but it appears that in mutant R193C the rcsF mutant is much wider (as pointed out in the previous review). Can the authors discuss this. It maybe that not enough cells were imaged as the error bar for width is quite large.

We have included more cells in our analyses (see change in EV2D legend). With the new dataset, the mutant R193C strain had cell width of $1.7 \pm 0.2 \mu\text{m}$, and R193C $\Delta rcsF$ had cell width of $1.8 \pm 0.3 \mu\text{m}$. The two strains have comparable widths.

The authors are right that putting rcsF+7 into the fat mutants would be hard to interpret, however the authors should make a rcsF-linker strain that reduces the length of the linker and introduce this into some of the fatter cells. Does this return the cells to WT expression levels now that the distance between rcsF and lgaA is reduced.

Unfortunately this is an experiment that cannot be performed for other reasons. It is impossible to shorten the linker of RcsF without impairing RcsF localization to the outer membrane (PMID: 34326538). This impairment of proper localization and retainment in the inner membrane leads to constitutive Rcs activation.

Minor Issues:

Line 134: mean cell width increased in an A22 dose-dependent manner (Fig. EV1A). There is no width measurement in EV1A, it is a graph showing Bgal v time in A22.

Thanks for noticing, we have corrected this part to refer only to Rcs activity.

EV2A: statistical difference between rcsF and WT are needed as the error bars appear to overlap at A22 2 ug/ml. See above comment about cell numbers and replicates.

We have included a statistical test and WT indeed had larger changes in width/length compared to $\Delta rcsF$. We added this information in the legend of Fig EV2A.

The authors reference Grinnell et al. which showed that cells will continue to grow in the presence of A22 in the conditions the authors tested in this study (line 136-137). The authors then state that the rcsF cells begin to die in the EV2 legend. Where these grown in the same manner as stated in the materials and methods for fig 1 (line 350). Please include this information for all experiments where A22 is used.

The protocols are indeed the same for both experiments. However, the data in EV2 are shown at 2.5 h, which corresponds to a much longer treatment. We have explicitly stated this difference in Lines 143-144.

Referee #3:

The revised manuscript provides evidence for a strong positive correlation of cell width with Rcs induction, and a negative correlation with periplasmic thickness. The revision improves the presentation significantly. This is an interesting and potentially important observation, and suggest that components of the Rcs regulon may play roles in cell shape regulation (in part investigated in a second study by this group).

We thank the reviewer for the positive comments!

Editor accepted the revised manuscript.